# Unifying Value Alignment and Assignment in Cross-Domain Offline Reinforcement Learning with Heterogeneous Datasets

**Zhongjian Qiao** [1]  **Jiafei Lyu** [2]  **Chenjia Bai** [3]  **Peisong Wang** [4]  **Siyang Gao** [1]  **Shuang Qiu** [†1]

## Abstract

Cross-domain offline reinforcement learning (RL) aims to learn a policy in the target domain with a limited target domain dataset and a source domain dataset that exhibits a dynamics shift. Training directly on the original source dataset typically leads to performance collapse. Recent studies perform data filtering from the perspective of dynamics alignment or value alignment to enable efficient policy transfer. However, these studies are typically validated on single-domain or single-behavior-policy source datasets. In this work, we explore a more general *heterogeneous* cross-domain offline RL setting, where the source datasets may be collected from multiple source domains by diverse behavior policies. We first uncover a critical yet overlooked issue in this setting: *value misassignment*. Empirically and theoretically, we demonstrate that value misassignment can undermine value alignment, mislead data filtering toward selecting suboptimal samples, and loosen the suboptimality gap, thereby degrading the agent's performance. To address this issue, we propose V2A, which integrates dynamics alignment, value alignment, and value assignment. V2A first employs temporally-consistent modality representation learning to extract dynamics modalities from the source dataset, followed by modality-aware advantage learning to rectify value alignment. Finally, it adopts a data filtering paradigm to selectively share source data for policy learning. Empirical results show that V2A significantly outperforms strong baseline methods under general heterogeneous cross-domain offline RL settings. Code is available at https://github.com/zq2r/V2A.git.

[1]City University of Hong Kong [2]Tencent [3]Institute of Artificial Intelligence (TeleAI), China Telecom [4]Institute of Automation, Chinese Academy of Sciences. [†]Corresponding Author. Correspondence to: Shuang Qiu <shuanqiu@cityu.edu.hk>.

*Proceedings of the 43$^{rd}$ International Conference on Machine Learning*, Seoul, South Korea. PMLR 306, 2026. Copyright 2026 by the author(s).

## 1. Introduction

Reinforcement learning (RL) (Sutton et al., 1998) has progressed remarkably in fields like embodied AI (Intelligence et al., 2025; Amin et al., 2025; Jiao et al., 2025) and large language models (Rafailov et al., 2023; Guo et al., 2025a). However, online RL requires intense environmental interactions, which can be expensive and risky in domains like healthcare. Offline RL (Levine et al., 2020; Prudencio et al., 2023) instead learns from pre-collected datasets, thereby eliminating online interaction costs. Yet, in many real-world scenarios, only a limited amount of target domain data is available, which can substantially limit the performance of offline RL. To address this challenge, recent studies (Liu et al., 2024; Wen et al., 2024; Lyu et al., 2025) explore Cross-Domain Offline RL, which leverages both a limited target dataset and an abundant dataset from a distinct but similar source domain to facilitate target policy learning.

Although it is promising to incorporate additional source domain data to mitigate the data scarcity issue, directly mixing the source and target domain datasets for training would degrade the policy performance in the target domain. The main reason is that the source and target domains may share different transition dynamics (Wen et al., 2024; Lyu et al., 2025), leading to the out-of-distribution (OOD) dynamics issue (Liu et al., 2024). To address this, recent works such as IGDF (Wen et al., 2024) and OTDF (Lyu et al., 2025) introduce dynamics-aligned data filtering, which measures the dynamics discrepancy between source and target domains and selectively shares source samples accordingly. However, these studies do not explicitly account for the optimality of source samples, which may result in suboptimal performance when source datasets comprise trajectories of mixed quality. More recently, DVDF (Qiao et al., 2025b) introduces the concept of *value alignment* to ensure that the selected samples are not only dynamics-consistent but also of high quality, making it more suitable for mixed-quality source datasets. Nevertheless, these studies have primarily validated their effectiveness within the setting where the source dataset is drawn from a single environment. In practical applications, such as robotic learning, source datasets may not only contain data of mixed quality but also be aggregated from multiple environments with diverse phys-

*Table 1.* Comparison between V2A and recent cross-domain offline RL baselines.

| | Dynamics Alignment | Value Alignment | Value Assignment | Heterogeneous Dataset |
|---|:---:|:---:|:---:|:---:|
| IGDF (Wen et al., 2024) | ✓ | ✗ | ✗ | ✗ |
| OTDF (Lyu et al., 2025) | ✓ | ✗ | ✗ | ✗ |
| DVDF (Qiao et al., 2025b) | ✓ | ✓ | ✗ | ✗ |
| **V2A (Ours)** | ✓ | ✓ | ✓ | ✓ |

ical parameters, highlighting a fundamental gap between previous studies and this realistic setting.

In this paper, we explore a more general cross-domain offline RL setting, where source datasets could originate from multiple source domains with distinct transition dynamics and are collected with diverse behavior policies, which we refer to as *heterogeneous* cross-domain offline RL. In this setting, we first uncover a critical yet previously overlooked issue: *value misassignment*. We further provide empirical and theoretical insights that value misassignment can undermine the effect of value alignment, mislead data filtering toward selecting suboptimal source data, and loosen the suboptimality gap in the target domain, thereby degrading the agent's performance in the target domain.

To resolve this challenge, we propose V2A, a simple yet effective framework that unifies **V**alue **A**lignment and **A**ssignment in heterogeneous cross-domain offline RL, going well beyond DVDF that focuses solely on value alignment. Our key insight is that mitigating value misassignment requires disentangling dynamics modalities within the source dataset. V2A first learns temporally-consistent modality representations in an Expectation-Maximization (EM) manner (Dempster et al., 1977) to infer the underlying dynamics for each source trajectory. We then relabel the source dataset and propose modality-aware advantage learning, ensuring advantage values reflect the true sample optimality under specific dynamics. Finally, we leverage data filtering to selectively share source samples for target policy learning. We present the major differences between V2A and recent cross-domain offline RL baselines in Table 1. Our main contributions are summarized as follows:

- We explore a more general but underexplored heterogeneous cross-domain offline RL setting and, within this setting, identify a critical yet previously overlooked value misassignment issue.

- We propose V2A that mitigates value misassignment with temporally-consistent modality representation learning and modality-aware advantage learning. V2A effectively integrates dynamics alignment, value alignment, and value assignment in a unified framework.

- We evaluate V2A on extensive cross-domain RL tasks with heterogeneous source datasets. Empirical results demonstrate that V2A significantly outperforms strong baselines across multiple tasks and datasets.

## 2. Preliminaries

**Reinforcement Learning.** We model a RL problem as a Markov Decision Process (MDP) (Puterman, 1990) defined by the six-tuple $\mathcal{M} = (\mathcal{S}, \mathcal{A}, P, r, \rho, \gamma)$ where $\mathcal{S}$ denotes the state space, $\mathcal{A}$ is the action space, $P : \mathcal{S} \times \mathcal{A} \to \Delta(\mathcal{S})$ represents the transition dynamics, $\Delta(\cdot)$ is the probability simplex, $r(s, a) : \mathcal{S} \times \mathcal{A} \to [-r_{\max}, r_{\max}]$ is the reward function, $\rho$ is the initial state distribution, and $\gamma \in [0, 1)$ is the discount factor. The objective of RL is to learn a policy $\pi : \mathcal{S} \to \Delta(\mathcal{A})$ that maximizes the expected discounted cumulative return $J_{\mathcal{M}}(\pi) := \mathbb{E}_\pi[\sum_{t=0}^{\infty} \gamma^t r(s_t, a_t)]$.

**Cross-Domain RL.** In cross-domain RL, we assume that we have access to a *source domain* MDP as $\mathcal{M}_{\mathrm{src}} = (\mathcal{S}, \mathcal{A}, P_{\mathrm{src}}, r, \rho, \gamma)$, and a *target domain* MDP as $\mathcal{M}_{\mathrm{tar}} = (\mathcal{S}, \mathcal{A}, P_{\mathrm{tar}}, r, \rho, \gamma)$, where the two MDPs only differ in their transition dynamics and share the identical state and action space, as widely studied in prior works (Wen et al., 2024; Lyu et al., 2025). Traditionally, $\mathcal{M}_{\mathrm{src}}$ represents a source domain with a single transition model. Beyond the prior setting, our work allows $\mathcal{M}_{\mathrm{src}}$ to be a mixed source domain with diverse transition dynamics. In the offline setting, we are only accessible to a source domain dataset $\mathcal{D}_{\mathrm{src}} = \{(s_i, a_i, r_i, s_{i+1})\}_{i=1}^{N_1}$ and a target domain dataset $\mathcal{D}_{\mathrm{tar}} = \{(s_i, a_i, r_i, s_{i+1})\}_{i=1}^{N_2}$, where $N_1 \gg N_2$. Cross-domain offline RL aims to leverage $\mathcal{D}_{\mathrm{src}} \cup \mathcal{D}_{\mathrm{tar}}$ to learn a well-performing policy in the target domain.

**Data Filtering for Cross-Domain Offline RL.** To address the OOD dynamics issue when directly using $\mathcal{D}_{\mathrm{src}} \cup \mathcal{D}_{\mathrm{tar}}$ for training, several studies (Wen et al., 2024; Lyu et al., 2025) explore dynamics-aligned data filtering, where a *score function* $h(s, a, s')$ is utilized to measure the dynamics alignment. IGDF (Wen et al., 2024) and OTDF (Lyu et al., 2025) obtain $h(\cdot)$ with contrastive learning and optimal transport, respectively. Furthermore, the recent work DVDF (Qiao et al., 2025b) proposes the concept of value alignment, which considers the optimality of source samples, raising the question of *whether it can handle mixed-quality source datasets*. DVDF defines a new score function $g(\cdot)$ as

$$g(s, a, s') = \lambda \cdot h(s, a, s') + (1 - \lambda) \cdot A^\star_{\mathrm{insrc}}(s, a), \quad (1)$$

where $\lambda$ is the trade-off coefficient, $A^\star_{\mathrm{insrc}}$ is the in-sample (Kostrikov et al., 2021) optimal advantage function on $\mathcal{D}_{\mathrm{src}}$, and is approximated by an advantage function pre-trained on $\mathcal{D}_{\mathrm{src}}$ with Sparse-QL (Xu et al., 2023a) algorithm.

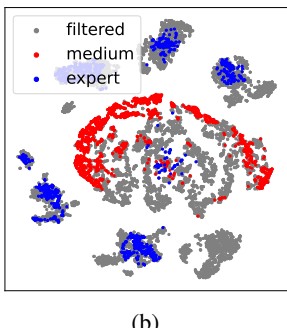 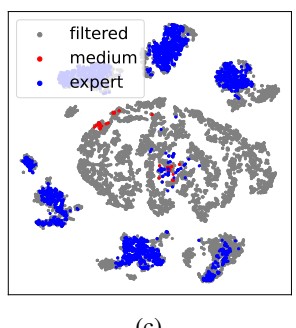 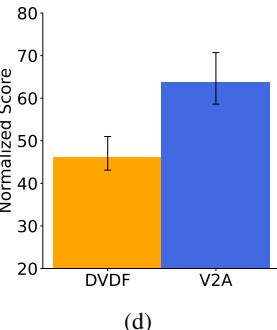

(a)      (b)      (c)      (d)

*Figure 1.* **(a)** Source dataset visualization. **(b)** Source data filtering visualization of DVDF. **(c)** Source data filtering visualization of V2A. **(d)** Performance comparison for DVDF and V2A on the target domain.

## 3. Motivating Example

This section uses an example to demonstrate that *DVDF may select **suboptimal** source domain data in the heterogeneous setting, thereby **hindering** effective target policy learning.*

**Experimental Setup.** We consider the following heterogeneous cross-domain offline RL scenario. We take the `halfcheetah-v2` task from MuJoCo (Todorov et al., 2012) as the target domain, and introduce medium-level kinematic shifts to `halfcheetah-v2` by modifying the corresponding `.xml` file to create the source domain, as detailed in Appendix D. For the target domain dataset, we sample 10% data from the `halfcheetah-medium-v2` dataset in D4RL. For the source domain dataset, we construct a mixed dataset consisting of: (1) 0.5M expert-level data sampled from the target domain, and (2) 0.5M medium-level data sampled from the source domain with dynamics shifts. We visualize the source domain dataset with t-SNE (Maaten & Hinton, 2008) in Figure 1 (a), where blue points represent expert data and red points represent medium data. We implement DVDF and V2A based on IGDF, and select 25% source domain data for policy learning.

**Results and Analysis.** We first visualize the data filtering results of DVDF and V2A, as shown in Figure 1 (b) and (c), respectively. In the visualizations, gray points represent filtered samples, blue points indicate selected expert data, and red points denote selected medium data. Recall that DVDF aims to select source data that exhibits low dynamics shift and high quality; consequently, it is expected to select exclusively expert data from the target domain. However, Figure 1 (b) reveals that DVDF retains a significant portion of medium-quality source data, which are suboptimal samples for policy learning. In contrast, Figure 1 (c) demonstrates that V2A predominantly selects expert data, aligning more closely with the desired outcome. Moreover, we evaluate DVDF and V2A on the target domain and present the performance comparison in Figure 1 (d). We observe that V2A achieves a normalized score of **63**, compared to DVDF's

score of 46, representing a **37%** performance improvement. We attribute this discrepancy to the value misassignment issue, which will be discussed in the following section.

## 4. Problem Setup and Value Misassignment

In this section, we formally formulate our problem settings. In Section 4.1, we demonstrate the setting and challenge of heterogeneous cross-domain offline RL. Subsequently, in Section 4.2, we systematically investigate and identify the issue of value misassignment within this setting, uncovering a fundamental limitation in prior work.

### 4.1. Heterogeneous Cross-Domain Offline RL

For a source domain dataset $\mathcal{D}_{\mathrm{src}}$ consisting of trajectories collected under the behavior policy $\mu_{\mathrm{src}}$ and dynamics $P_{\mathrm{src}}$, we first present the default cross-domain offline RL setting with unimodal behavior policy and dynamics.

**Definition 4.1** (Unimodal source dataset). We say the source dataset behavior policy $\mu_{\mathrm{src}}$ and source domain dynamics $P_{\mathrm{src}}$ are unimodal when they follow:

$$\mu_{\mathrm{src}}(\cdot|s) \sim \mathcal{N}\left(m_1(s), \sigma_1^2(s)\right),$$
$$P_{\mathrm{src}}(\cdot|s, a) \sim \mathcal{N}\left(m_2(s, a), \sigma_2^2(s, a)\right),$$

where $\mathcal{N}(m, \sigma^2)$ denotes the Gaussian distribution with mean $m$ and variance $\sigma^2$.

This setting assumes the source domain dataset is collected in a single domain with a single behavior policy, which is overly strict and often impractical. This work considers a more general heterogeneous cross-domain offline RL setting where the source dataset is heterogeneous, originating from multiple domains via diverse behavior policies. Thus $\mu_{\mathrm{src}}$ and $P_{\mathrm{src}}$ can be characterized by multi-modal distributions. Inspired by recent works (Wang et al., 2024a) in single-domain offline RL, we can define the heterogeneous source dataset via a Gaussian Mixture Model.

**Definition 4.2** (Heterogeneous source dataset). A source dataset $\mathcal{D}_{\mathrm{src}}$ is heterogeneous if it can be divided into $n$

sub-datasets $\{\mathcal{D}_{\text{src}}^i\}_{i=1}^n$ with unimodal behavior policy and dynamics $\{\mu_{\text{src}}^i, P_{\text{src}}^i\}_{i=1}^n$. Then $\mu_{\text{src}}$ and $P_{\text{src}}$ can be expressed as Gaussian mixture models with

$$\mu_{\text{src}}(\cdot|s) = \sum_{i=1}^n \alpha_i \cdot \mu_{\text{src}}^i(\cdot|s),$$
$$P_{\text{src}}(\cdot|s,a) = \sum_{i=1}^n \alpha_i \cdot P_{\text{src}}^i(\cdot|s,a),$$

where $\left\{\alpha_i = \frac{|\mathcal{D}_{\text{src}}^i|}{|\mathcal{D}_{\text{src}}|}\right\}_{i=1}^n$ are the mixing coefficients for each sub-dataset, and $\sum_{i=1}^n \alpha_i = 1$.

Recent studies in cross-domain offline RL, such as IGDF (Wen et al., 2024) and OTDF (Lyu et al., 2025), rely exclusively on dynamics-aligned data filtering. Consequently, they overlook the varying quality of source data, rendering them ineffective for datasets induced by multimodal behavior policies. In contrast, DVDF (Qiao et al., 2025b) incorporates both dynamics and value alignment, ostensibly making it better suited for heterogeneous settings. However, as shown in Section 3, the performance of DVDF remains limited in such scenarios. We attribute this shortfall to a critical yet overlooked phenomenon: *value misassignment*. This issue undermines the value alignment DVDF depends on, as elaborated in the following subsection.

## 4.2. Value Misassignment in Cross-Domain Offline RL

In this section, we formally identify the value misassignment issue in heterogeneous cross-domain offline RL. We first present the concept of dynamics misalignment commonly considered in previous works (Xu et al., 2023b; Wen et al., 2024; Lyu et al., 2025).

**Definition 4.3** (Dynamics Misalignment). For the source domain dynamics $P_{\text{src}}(\cdot|s,a)$ and target domain dynamics $P_{\text{tar}}(\cdot|s,a)$, the *dynamics misalignment* between $P_{\text{src}}(\cdot|s,a)$ and $P_{\text{tar}}(\cdot|s,a)$ is measured as $\text{Dis}\left(P_{\text{src}}(\cdot|s,a), P_{\text{tar}}(\cdot|s,a)\right)$, where $\text{Dis}(\cdot)$ is the distribution distance measure, such as total variation $D_{\text{TV}}(\cdot)$.

In addition to dynamics misalignment, DVDF (Qiao et al., 2025b) proposes an orthogonal concept of value misalignment, which has been shown to be critical for cross-domain offline RL, both theoretically and empirically.

**Definition 4.4** (Value Misalignment). For a source domain $\mathcal{M}_{\text{src}}$ and source dataset $\mathcal{D}_{\text{src}}$ with behavior policy $\mu_{\text{src}}$, the *value misalignment* is defined as $\Delta J_{\mathcal{M}_{\text{src}}} := |J_{\mathcal{M}_{\text{src}}}(\mu_{\text{src}}) - J_{\mathcal{M}_{\text{src}}}(\pi_{\text{insrc}}^\star)|$, where $\pi_{\text{insrc}}^\star$ denotes the in-sample optimal policy on $\mathcal{D}_{\text{src}}$. Intuitively, $\Delta J_{\mathcal{M}_{\text{src}}}$ measures the suboptimality gap of $\mu_{\text{src}}$ on the source domain.

Before introducing the concept of value misassignment, we first analyze the suboptimality gap of a policy $\hat{\pi}$ trained on the heterogeneous source dataset $\mathcal{D}_{\text{src}}$ and evaluated on the target domain $\mathcal{M}_{\text{tar}}$. Specifically, we denote the

suboptimality gap of $\hat{\pi}$ on the target domain as

$$\text{SubOpt}(\hat{\pi}) := J_{\mathcal{M}_{\text{tar}}}(\hat{\pi}) - J_{\mathcal{M}_{\text{tar}}}(\pi_{\text{tar}}^\star),$$

where $\pi_{\text{tar}}^\star$ is the optimal policy in the target domain.

**Proposition 4.5** (Suboptimality gap with heterogeneous source datasets). *Denote the MDP of the target domain and multiple source domains as $\mathcal{M}_{\text{tar}}$ and $\{\mathcal{M}_{\text{src}}^i\}_{i=1}^n$. For a policy $\hat{\pi}$ trained on $\mathcal{D}_{\text{src}}$ defined in Definition 4.2, under some mild assumptions, the suboptimality gap on $\mathcal{M}_{\text{tar}}$ can be bounded as*

$$\text{SubOpt}(\hat{\pi}) \leq \sum_{i=1}^n \alpha_i \cdot \Big( \underbrace{|J_{\mathcal{M}_{\text{src}}^i}(\mu_{\text{src}}^i) - J_{\mathcal{M}_{\text{src}}^i}(\pi_{\text{insrc}}^{i\star})|}_{\text{value misalignment}}$$
$$+ C \cdot \underbrace{\sup_{s,a} \left[ D_{\text{TV}}(P_{\text{src}}^i(\cdot|s,a), P_{\text{tar}}(\cdot|s,a)) \right]}_{\text{dynamics misalignment}} \Big) + \Delta,$$

*where we define $\Delta := \sum_{i=1}^n \alpha_i \left( \epsilon_{\text{insrc}}^{i\star} + \epsilon_{\mu_{\text{src}}}^i \right)$, $\epsilon_{\text{insrc}}^{i\star} := |J_{\mathcal{M}_{\text{src}}^i}(\pi_{\text{insrc}}^{i\star}) - J_{\mathcal{M}_{\text{src}}^i}(\pi_{\text{src}}^\star)|$, and $\epsilon_{\mu_{\text{src}}}^i := |J_{\mathcal{M}_{\text{src}}^i}(\hat{\pi}) - J_{\mathcal{M}_{\text{src}}^i}(\mu_{\text{src}}^i)|$. Here $\pi_{\text{insrc}}^{i\star}$ is the in-sample optimal policy on $\mathcal{D}_{\text{src}}^i$, $\pi_{\text{src}}^{i\star}$ is the optimal policy in $\mathcal{M}_{\text{src}}^i$, $C$ is a constant, and $\{\alpha_i\}_{i=1}^n$ are the mixing coefficients for sub-datasets.*

This shows that $\text{SubOpt}(\hat{\pi})$ can be bounded by a weighted sum of per-source-domain value and dynamics misalignment terms with weights $\alpha_i$ and an additional estimation error term $\Delta$, shown bounded in Appendix B, that aggregates finite-sample and approximation errors. *This proposition inspires us to increase the weights of sub-datasets with smaller value and dynamics misalignment terms to obtain a tighter sup-optimality gap.* In other words, we can prioritize samples from these sub-datasets during data filtering. According to performance difference lemma (Kakade & Langford, 2002), the value misalignment term can be approximated by the average optimal advantage value: $\mathbb{E}_{(s,a)\sim\mathcal{D}_{\text{src}}^i}[A_{\text{insrc}}^{i\star}(s,a)] \approx J_{\mathcal{M}_{\text{src}}^i}(\mu_{\text{src}}^i) - J_{\mathcal{M}_{\text{src}}^i}(\pi_{\text{insrc}}^{i\star})$, where $A_{\text{insrc}}^{i\star}(\cdot)$ denotes the in-sample optimal advantage function on $\mathcal{D}_{\text{src}}^i$. Recall that DVDF pre-trains an advantage function on the whole source dataset to approximate $A_{\text{insrc}}^\star(\cdot)$. Since $\mathbb{E}_{(s,a)\sim\mathcal{D}_{\text{src}}^i}[A_{\text{insrc}}^\star(s,a)] \approx J_{\mathcal{M}_{\text{src}}}(\mu_{\text{src}}^i) - J_{\mathcal{M}_{\text{src}}}(\pi_{\text{insrc}}^\star)$ where $\pi_{\text{insrc}}^\star$ denotes the in-sample optimal policy on $\mathcal{D}_{\text{src}}$, DVDF implicitly measures the discrepancy between $\mu_{\text{src}}^i$ and $\pi_{\text{insrc}}^\star$ on $\mathcal{M}_{\text{src}}$, inconsistent with the value misalignment term in Proposition 4.5. We define this inconsistency as *value misassignment*:

**Definition 4.6** (Value Misassignment). For the heterogeneous source dataset $\mathcal{D}_{\text{src}}$, there exist some sub-datasets $\{\mathcal{D}_{\text{src}}^j\}_{j=1}^m$ such that $\mathbb{E}_{(s,a)\sim\mathcal{D}_{\text{src}}^j}[A_{\text{insrc}}^{j\star}(s,a)] \neq \mathbb{E}_{(s,a)\sim\mathcal{D}_{\text{src}}^j}[A_{\text{insrc}}^\star(s,a)]$, $j = 1, 2, ..., m$.

Intuitively, value misassignment implies assigning incorrect advantage values to source samples, leading to an inaccurate assessment of sample optimality. This may mislead data filtering, thus loosening the suboptimality gap. For example,

consider a source sub-dataset $\mathcal{D}_{\mathrm{src}}^i$, whose behavior policy is the in-sample optimal policy: $\mu_{\mathrm{src}}^i = \pi_{\mathrm{insrc}}^{i\star}$. In this case, $\mathbb{E}_{(s,a)\sim\mathcal{D}_{\mathrm{src}}^i}\left[A_{\mathrm{insrc}}^{i\star}(s,a)\right] \approx 0$. However, $\mu_{\mathrm{src}}^i$ might be sub-optimal in $\mathcal{M}_{\mathrm{src}}$, i.e., $\mathbb{E}_{(s,a)\sim\mathcal{D}_{\mathrm{src}}^i}\left[A_{\mathrm{insrc}}^{\star}(s,a)\right] < 0$. This underestimates the optimality of samples in $\mathcal{D}_{\mathrm{src}}^i$, causing them to be deprioritized during data filtering. Conversely, weights of suboptimal samples may increase, thereby loosening the suboptimality gap.

## 5. Methodology

In Section 4.2, we have revealed the issue of value misassignment in heterogeneous cross-domain offline RL. In this section, we present our solution to this issue, V2A. It integrates dynamics alignment, value alignment, and value assignment into a unified framework, incorporating temporally-consistent modality representation learning, modality-aware advantage learning, and a data filtering paradigm to selectively share source data for policy learning.

### 5.1. Temporally-Consistent Modality Representation Learning

The first step of V2A is to extract different dynamics modalities from the source dataset. Typically, we can maximize the Evidence Lower Bound (ELBO) for the dynamics log-likelihood $\mathbb{E}_{\mathcal{D}_{\mathrm{src}}}\left[\log p_\theta(s'|s,a)\right]$, which is defined as

$$\mathsf{ELBO}_{\psi,\theta} := \mathbb{E}_{(s,a,s')\sim\mathcal{D}_{\mathrm{src}},z\sim q_\psi(\cdot|s,a,s')} \qquad (2)$$
$$\left[\log p_\theta(s'|s,a,z) - D_{\mathrm{KL}}(q_\psi(\cdot|s,a,s'),p(\cdot))\right],$$

where $q_\psi$, $p_\theta$ are the representation encoder and dynamics decoder, $z$ is the dynamics modality representation, and $p(z)$ denotes the prior distribution of $z$ setting to be $\mathcal{N}(0,I)$. By optimizing Equation 2, we learn a dynamics modality representation $z$ for each transition $(s,a,s')$ in $\mathcal{D}_{\mathrm{src}}$. However, we demonstrate that such an ELBO can introduce temporal inconsistency. Specifically, for transitions within a trajectory, they follow the same transition dynamics and should share identical dynamics representations. However, in Equation 2, $q_\psi$ encodes a unique representation $z$ for each transition $(s,a,s')$ even when they are sampled from the same trajectory, resulting in temporal inconsistency for the learned representation. To address this issue, we propose Temporally-Consistent ELBO, which encodes the dynamics modality representation at the trajectory level while decoding the transition dynamics at the step level.

**Proposition 5.1** (Temporally-Consistent ELBO). *Letting* $\mathcal{D}_{\mathrm{src}}$ *be the source domain dataset of trajectories* $\tau = (s_0,a_0,s_1,\ldots,s_T)$, *under the Markov property assumption for transition dynamics, the temporally-consistent ELBO for the dynamics log-likelihood* $\mathbb{E}_{\tau\sim\mathcal{D}_{\mathrm{src}}}[\log p_\theta(\tau)]$ *is*

$$\mathsf{TC\text{-}ELBO}_{\psi,\theta} := \mathbb{E}_{\tau\sim\mathcal{D}_{\mathrm{src}},z\sim q_\psi(\cdot|\tau)} \qquad (3)$$
$$\left[\sum_{t=0}^{T-1}\log p_\theta(s_{t+1}\mid s_t,a_t,z) - D_{\mathrm{KL}}\big(q_\psi(\cdot\mid\tau),p(\cdot)\big)\right],$$

where $q_\psi$ is the trajectory-level representation encoder, $p_\theta$ is the transition-level dynamics decoder, $p(z) = \mathcal{N}(0,I)$ is the prior over trajectory dynamics representation $z$.

Equation 3 enforces temporal consistency by sharing a single $z$ across all transitions in one trajectory $\tau$. We next describe the construction of $q_\psi$ and $p_\theta$. The encoder $p_\psi$ needs to learn the dynamics modality representation of trajectories with variable lengths. Inspired by previous works (Nagabandi et al., 2018; Chen et al., 2021b; 2024) which utilize Recurrent Neural Network (RNN) (Hochreiter & Schmidhuber, 1997) as a representation extractor to map the input trajectories to some task-specific meta-parameters, we use a similar RNN structure to learn and infer the dynamics representation of trajectories. Formally, let $\tau = (s_0,a_0,s_1,...,s_T)$ be a trajectory sampled from the source dataset where $T$ denotes the trajectory horizon, we feed $\tau$ into a RNN $q_\psi(\tau)$ to output a latent representation distribution $z \sim \mathcal{N}(m(\tau),\sigma^2(\tau))$ at the final step. For $p_\theta$, we follow previous model-based RL literature (Janner et al., 2019; Yu et al., 2020), and build it as an ensemble fully-connected dynamics model. At each step $t$, transition $(s_t,a_t,s_{t+1})$ and the latent representation $z$ are fed into $p_\theta(s_{t+1}|s_t,a_t,z)$ to compute the likelihood. Then, TC-ELBO$_{\psi,\theta}$ in Equation 3 is maximized by alternatively optimizing $q_\psi$ and $p_\theta$ via updating $\psi$ and $\theta$, in an Expectation-Maximization (EM) manner. Specifically, at the $k$-th iteration of the EM updates:

**E-Step.** The E-step fixes $p_{\theta^{(k)}}$ and optimizes $q_\psi$ via

$$\psi^{(k)} = \arg\max_\psi \mathbb{E}_{\tau\sim\mathcal{D}_{\mathrm{src}},z\sim q_\psi(\cdot|\tau)} \qquad (4)$$
$$\left[\sum_{t=0}^{T-1}\log p_{\theta^{(k)}}(s_{t+1}|s_t,a_t,z) - D_{\mathrm{KL}}\big(q_\psi(\cdot|\tau),p(\cdot)\big)\right],$$

which intuitively encourages the inferred latent representations $z$ to capture and better explain the observed transition dynamics for each trajectory.

**M-Step.** The M-step fixes $q_{\psi^{(k)}}$ and optimizes $p_\theta$ via

$$\theta^{(k+1)} = \arg\max_\theta \mathbb{E}_{\tau\sim\mathcal{D}_{\mathrm{src}},z\sim q_{\psi^{(k)}}(\cdot|\tau)} \qquad (5)$$
$$\left[\sum_{t=0}^{T-1}\log p_\theta(s_{t+1}|s_t,a_t,z)\right],$$

where the term $-D_{\mathrm{KL}}\big(q_{\psi^{(k)}}(\cdot|\tau),p(\cdot)\big)$ is dropped as $q_{\psi^{(k)}}$ is fixed. In essence, the M-step is supervised learning on trajectories with the inferred latent representations $z$.

### 5.2. Modality-Aware Advantage Learning for Unifying Value Alignment and Assignment

By iterating through the EM steps until convergence, we obtain the dynamics modality representation $z$ for all the trajectories in the source dataset. In this section, we leverage the learned representation to tackle the value misassignment issue in heterogeneous cross-domain offline RL.

According to the analysis in Section 4.2, the primary cause of the value misassignment issue in DVDF is that, it pretrains an advantage function $A(s, a)$ directly on $\mathcal{D}_{\text{src}}$, which consequently renders $A(s, a)$ unable to distinguish between different dynamics $\{P_{\text{src}}^i\}_{i=1}^n$. To address this, we first relabel $\mathcal{D}_{\text{src}}$ to incorporate the learned dynamics representation $z$ into the transitions. This yields a relabeled source dataset $\mathcal{D}'_{\text{src}} = \{(s, a, r, s', z)\}$, where $z \sim p(\tau)$ is the inferred dynamic representation for each trajectory. Then we propose modality-aware advantage learning. We incorporate the learned representation $z$ into the advantage pretraining process, which enables the advantage function to distinguish between different dynamics modalities and to properly assign advantage values. Specifically, we take $z$ as part of the input of $Q$-function and $V$-function, and update $Q$ and $V$ in the Sparse-QL scheme as follows:

$$\mathcal{L}_V = \mathbb{E}_{(s,a,z)\sim\mathcal{D}'_{\text{src}}}\left[\mathbb{I}\left(1 + \frac{Q(s,a,z) - V(s,z)}{2\alpha} > 0\right) \cdot \quad (6)\right.$$
$$\left.\left(1 + \frac{Q(s,a,z) - V(s,z)}{2\alpha}\right)^2 + \frac{V(s,z)}{\alpha}\right],$$

$$\mathcal{L}_Q = \mathbb{E}_{(s,a,r,s',z)\sim\mathcal{D}'_{\text{src}}}\left[\left(r + \gamma V(s',z) - Q(s,a,z)\right)^2\right], \quad (7)$$

where $\alpha$ is the regularization coefficient. Since we only need to obtain the advantage function, we do not involve the policy learning process. The modality-aware advantage function is derived as $A(s, a, z) = Q(s, a, z) - V(s, z)$. Compared with the typical advantage function $A(s, a)$ as adopted in DVDF, $A(s, a, z)$ offers more modality-related information with $z$. Intuitively, $A(s, a, z)$ assigns different advantage values to state-action pairs $(s, a)$ with varying dynamics modalities, which unifies value alignment and value assignment. To enable dynamics alignment, we define a score function $f(s, a, s', z)$ similar to $g(s, a, s')$ in DVDF:

$$f(s, a, s', z) = \lambda \cdot h(s, a, s') + (1-\lambda) \cdot \text{Norm}(A(s, a, z)), \quad (8)$$

where $h(s, a, s')$ measures the dynamics alignment of the transition $(s, a, s')$ with the target dynamics and is learned with either IGDF or OTDF, $\lambda$ is the trade-off hyperparameter, and $\text{Norm}(\cdot)$ is the normalization operator. $f(s, a, s', z)$ enables modality awareness with $z$, thus unifying dynamics alignment, value alignment, and value assignment. In contrast, $g(s, a, s')$ only considers value alignment and dynamics alignment, and overlooks the value misassignment issue. Then we adopt a data filtering paradigm following previous works (Wen et al., 2024; Lyu et al., 2025; Qiao et al., 2025b) and select the top $\xi$-quantile of batch source samples for training:

$$\mathcal{L}_Q(\phi) = \frac{1}{2}\mathbb{E}_{(s,a,s')\sim\mathcal{D}_{\text{tar}}}\left[(Q_\phi - \mathcal{T}Q_\phi)^2\right] + \quad (9)$$
$$\frac{1}{2}\mathbb{E}_{(s,a,s',z)\sim\mathcal{D}'_{\text{src}}}\left[w(s,a,s',z)f(s,a,s',z)(Q_\phi - \mathcal{T}Q_\phi)^2\right],$$

where $w(s, a, s', z) = \mathbb{I}(f(s, a, s', z) > f_{\xi\%})$ is an indicator function, and $f_{\xi\%}$ means the $\xi$-th quantile of the $f$-values

among source samples in a batch. Then we update the policy with IQL (Kostrikov et al., 2021). We present the pseudocode of V2A in Appendix E.2.

# 6. Experiments

In this section, we empirically examine the effectiveness of V2A and study its sensitivity to major hyperparameters. We aim to answer the following questions: **(a)** Can V2A outperform prior strong baselines with heterogeneous source datasets under various dynamics shifts and dataset qualities? **(b)** Does V2A learn reasonable modality representations?

## 6.1. Main Results with Heterogeneous Datasets

**Settings.** We leverage four tasks (halfcheetah, hopper, walker2d, ant) from OpenAI Gym (Brockman et al., 2016) as target domains. Since only limited target domain data can be accessed, we sample $10\%$ data from D4RL (Fu et al., 2020) MuJoCo datasets with various data qualities (medium, medium-expert, expert) as the target domain datasets. For the source domain with dynamics shifts, following previous works (Lyu et al., 2025; Qiao et al., 2025b), we consider two shift types: *kinematic shifts* and *morphology shifts*. For each shift type, we further introduce two shift levels: *easy* and *medium*. Accordingly, for each target domain, the corresponding source domains encompass four distinct dynamics (2[types] × 2[levels]). We present more details in Appendix D. For the source domain datasets, we consider heterogeneous datasets with multi-modal behavior policy and dynamics. Specifically, following a similar data collection procedure as D4RL, we collect medium-replay and medium-expert datasets in all source domains. For each task, we mix datasets of the same data quality collected from the four source domains in equal proportions (each for $25\%$). Consequently, each source domain dataset exhibits a multi-modal behavior policy (medium-replay or medium-expert) and transition dynamics (kinematic and morphology shifts with different shift levels). The source domain datasets contain nearly 1M samples, much more than that of target domain datasets.

**Baselines.** We compare our method V2A against six baselines: IQL (Kostrikov et al., 2021), BOSA (Liu et al., 2024), DARA (Liu et al., 2022a), IGDF (Wen et al., 2024), OTDF (Lyu et al., 2025), DVDF (Qiao et al., 2025b). Since DVDF and V2A are algorithmic frameworks rather than standalone algorithms, we combine them with IGDF and OTDF respectively for performance comparison.

**Results.** We run all methods for 1M gradient steps across 5 random seeds, and present the performance comparison results of V2A against baselines in Table 2. We report the normalized score performance in the target domain.

The empirical results clearly indicate that V2A brings more

*Table 2.* **Performance comparison between V2A and other baselines.** half=halfcheetah, m=medium, me=medium-expert, mr=medium-replay, e=expert. We report the normalized score evaluated in the target domain, and $\pm$ captures the standard deviation across 5 seeds. For each task, we **bold** and highlight the best scores in cyan.

| Source | Target | IQL | BOSA | DARA | IGDF | DVDF | V2A | OTDF | DVDF | V2A |
|---|---|---|---|---|---|---|---|---|---|---|
| half-mr | m | 27.9 | 14.3 | 20.2 | 24.6 | **28.3±1.8** | 26.8±2.4 | 27.3 | 24.9±1.5 | **35.6±1.9** |
| half-mr | me | 31.2 | 26.9 | 33.5 | 37.2 | 44.5±1.1 | **51.4±1.7** | 41.8 | 46.3±3.8 | **57.3±2.4** |
| half-mr | e | 22.4 | 17.8 | 28.7 | **28.2** | 25.0±1.2 | 26.4±0.5 | 20.6 | **35.2±1.8** | 30.7±2.3 |
| half-me | m | 38.7 | 41.4 | 45.2 | 42.6 | 52.9± 3.3 | **60.7±5.8** | 45.1 | 38.1±3.7 | **49.4±4.2** |
| half-me | me | 40.7 | 36.1 | 44.8 | 32.4 | 48.3±2.0 | **51.7±3.4** | 35.7 | 46.2±2.4 | **49.5±2.8** |
| half-me | e | 51.6 | 44.7 | 49.2 | 53.7 | 46.9±2.1 | **62.7±4.8** | 42.6 | 55.7±4.5 | **73.8±6.7** |
| hopper-mr | m | 53.1 | 47.6 | 48.8 | 45.3 | 50.6±1.7 | **58.8±2.2** | 56.1 | 48.5±2.4 | **58.3±1.8** |
| hopper-mr | me | 28.5 | 46.7 | **58.9** | 39.5 | 51.2±2.5 | **56.8±3.9** | 27.6 | 43.1±3.4 | **52.7±5.6** |
| hopper-mr | e | **66.3** | 24.7 | 42.1 | 53.7 | 46.8±3.1 | **61.7±4.8** | 62.7 | **71.3±2.6** | 64.6±5.4 |
| hopper-me | m | 62.5 | 40.4 | 55.2 | 46.3 | 52.8±4.1 | **56.4±2.3** | 54.4 | 58.2±3.7 | **65.7±2.5** |
| hopper-me | me | 21.3 | 54.9 | 37.3 | 65.4 | 66.8±4.2 | **76.5±2.8** | 48.3 | 55.4±3.8 | **74.3±4.6** |
| hopper-me | e | 45.1 | 58.8 | 56.3 | 58.6 | 55.3±4.4 | **64.8±3.6** | 67.7 | 48.5±6.2 | **72.4±3.5** |
| walker2d-mr | m | 14.6 | 8.9 | 18.3 | 11.6 | 23.4±1.0 | **42.7±5.1** | 31.4 | 29.6±1.5 | **35.1±4.6** |
| walker2d-mr | me | 32.9 | 21.4 | 31.6 | **35.2** | 30.0±1.4 | 31.4±2.7 | 26.9 | 34.9±2.3 | **40.6±1.4** |
| walker2d-mr | e | 37.1 | 17.8 | 42.3 | 34.3 | 36.2±2.7 | **52.3±1.9** | 43.0 | 49.8±3.0 | **63.1±7.5** |
| walker2d-me | m | 75.3 | 43.6 | 62.8 | 72.5 | 73.3±4.2 | **84.6±2.3** | 64.9 | 72.6±4.5 | **78.2±1.7** |
| walker2d-me | me | 59.7 | 62.4 | 65.5 | 64.9 | 71.7±2.3 | **76.6±2.7** | 70.2 | 73.4±1.6 | **82.6±2.9** |
| walker2d-me | e | 65.8 | 51.7 | 67.4 | 73.6 | 67.1±2.8 | **79.4±5.6** | 57.6 | 52.7±1.9 | **71.3±4.3** |
| ant-mr | m | 46.2 | 31.5 | 43.7 | 37.4 | 48.6±0.6 | **54.3±1.0** | 43.1 | 48.5±1.2 | **56.6±3.7** |
| ant-mr | me | 62.3 | 54.6 | 72.4 | 64.4 | 73.5±6.7 | **81.7±4.5** | 68.2 | 74.4±3.6 | **85.6±6.9** |
| ant-mr | e | 58.3 | 76.2 | 70.8 | **73.1** | 65.5±3.0 | 70.2±2.3 | **86.5** | 74.3±4.6 | 78.7±3.1 |
| ant-me | m | **98.3** | 84.4 | 80.6 | 78.2 | 88.7±4.8 | **96.3±2.5** | 83.7 | 87.4±3.8 | **93.3±2.0** |
| ant-me | me | 103.5 | 107.9 | 97.6 | 101.3 | 105.5±1.2 | **114.0±1.8** | 105.7 | 112.1±1.0 | **117.3±2.5** |
| ant-me | e | 106.4 | 113.1 | 108.2 | 112.7 | 121.8±0.5 | **124.3±0.2** | 108.4 | 114.8±0.4 | **126.2±0.8** |
| **Total Score** | | 1249.7 | 1127.8 | 1281.4 | 1286.7 | 1374.7 | **1562.5** | 1319.5 | 1395.9 | **1612.9** |

significant performance improvement on base algorithms (IGDF and OTDF) than DVDF with heterogeneous source datasets, while clearly surpassing other baselines (IQL, BOSA, DARA) across various tasks and dataset qualities. Notably, IGDF-based V2A outperforms IGDF and DVDF in **20** out of 24 tasks, and OTDF-based V2A achieves a higher score than OTDF and DVDF in **21** out of 24 tasks. In terms of the total score, V2A achieves performance improvements of **21.4%** (from 1286.7 to 1562.5) for IGDF and **22.2%** (from 1319.5 to 1612.9) for OTDF, outperforming DVDF that delivers improvements of only 6.8% (from 1286.7 to 1374.7) and 5.5% (from 1319.5 to 1395.9) for IGDF and OTDF. We attribute this discrepancy to the impact of value misassignment, which causes DVDF to select suboptimal samples, a key issue that V2A is designed to address.

### 6.2. Analysis of Learned Modality Representation

To answer question (b), we conduct an empirical study to verify whether the modality representations learned by V2A are reasonable. Ideally, trajectories sampled from the same source domain should be encoded into the same repre-

sentation space. We select the *halfcheetah-medium-replay* and *halfcheetah-medium-expert* datasets as source domain datasets, following the experimental setup in Section 6.1. Each dataset comprises four distinct dynamics, covering two shift types (*kinematic* and *morphology*) and two shift levels (*easy* and *medium*). Following the procedure outlined in Section 5.1, we train a representation encoder $q_\psi$ on each source domain dataset and learn a latent representation $z$ for each trajectory. For each type of dynamics in the dataset, we randomly sample ten trajectories and visualize their latent representations $z$ using t-SNE. The resulting visualization is shown in Figure 2 (a). We find that the representations of trajectories sampled from the same source domain or source domains with the same shift type are closer to each other, while the representations of trajectories from source domains with different shift types are farther apart. This indicates that the representations learned with $q_\psi$ are able to distinguish different dynamics modalities.

We further investigate the impact of $z$ on advantage learning. Specifically, we perform advantage learning with V2A and DVDF on the same source datasets and plot the probability

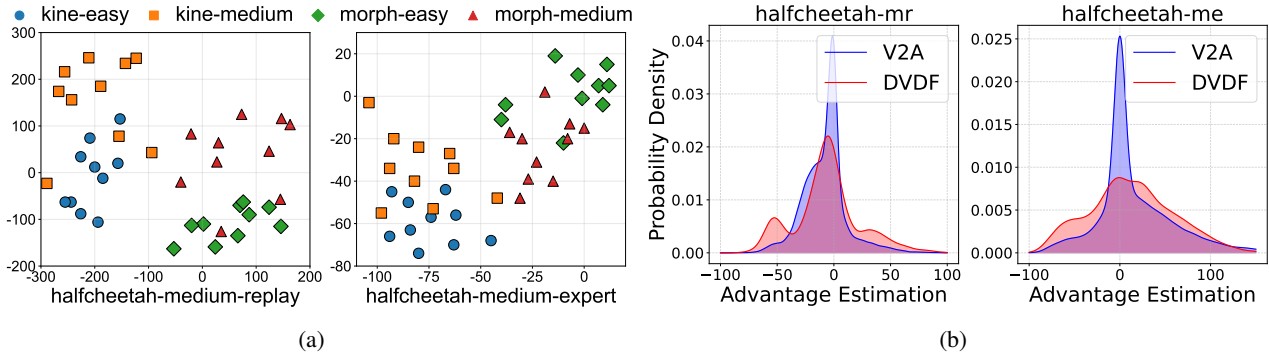

*Figure 2.* **(a)** Visualization of the learned modality representation. **(b)** Comparison of advantage distribution for V2A and DVDF.

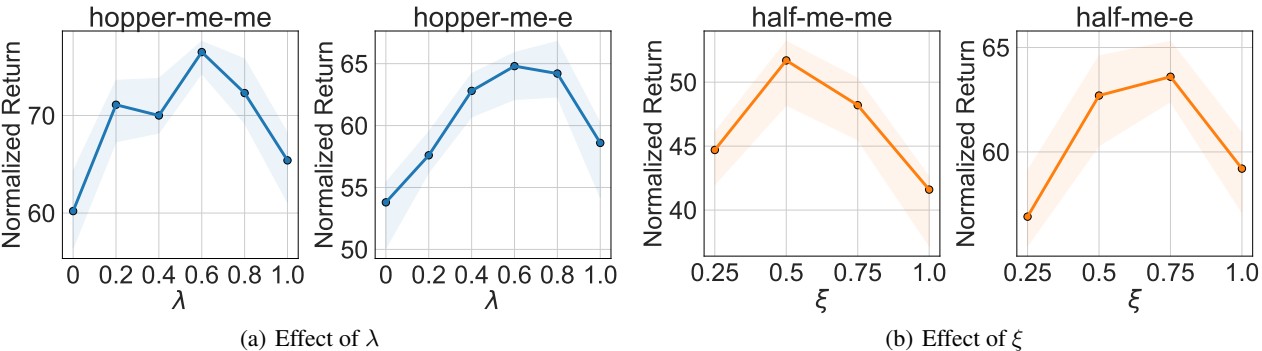

*Figure 3.* Parameter study on $\lambda$ and $\xi$. "me-e" means that the source dataset is medium-expert, the target dataset is expert, and so on.

density distributions of the advantage values in Figure 2 (b). We observe that the advantage distribution learned by V2A is sharper, which aligns with our expectation: since the data collected under different dynamics share the same data quality (*medium-expert* or *medium-replay*), they should exhibit similar optimality. In contrast, DVDF yields a flatter advantage distribution, indicating that it misestimates the optimality for certain source samples.

### 6.3. Parameter Study

In this section, we explore the impact of the main hyperparameters: the trade-off coefficient $\lambda$ and the data selection ratio $\xi$ on the performance of V2A. Each experiment is run with 1M gradient steps across 5 seeds.

**Trade-off coefficient** $\lambda$. A larger $\lambda$ emphasizes the importance of dynamics alignment, while a smaller $\lambda$ favors selecting higher-quality samples. Following the setup in Section 6.1, we use the *hopper-me* dataset with dynamics shifts as the source dataset, and select *hopper-me* and *hopper-e* as the target datasets, respectively. We vary $\lambda$ across $\{0, 0.2, 0.4, 0.6, 0.8, 1.0\}$ and present the experimental results in Figure 3 (a). We observe that performance degrades when $\lambda$ is either too small or too large, whereas a moderate value such as $0.6$ yields satisfactory performance.

Similar to DVDF, we fix $\lambda$ to a constant (i.e., $\lambda = 0.6$) in our experiments, without tuning it for each dataset individually.

**Data selection ratio** $\xi$. A larger $\xi$ selects more source data for training, and vice versa. To investigate the effect of $\xi$, we use the *halfcheetah-me* dataset with dynamics shifts as the source dataset, and select *halfcheetah-me* and *halfcheetah-e* as the target datasets, respectively. We vary $\xi$ across $\{0.25, 0.5, 0.75, 1.0\}$ and present the experimental results in Figure 3 (b). We find that setting $\xi$ to $0.5$ or $0.75$ yields good performance, and we simply fix $\xi$ to $0.5$ in all experiments.

## 7. Conclusion

In this work, we systematically investigate a novel and general heterogeneous cross-domain offline RL setting. We identify a critical value misassignment issue in this setting. Both empirically and theoretically, we demonstrate that value misassignment can mislead data filtering toward selecting suboptimal samples, thus hindering effective policy learning. To address this, we propose V2A, which performs temporally-consistent modality representation learning and modality-aware advantage learning, as well as a data filtering paradigm to unify value alignment and value assignment. Empirical results on various heterogeneous cross-domain offline RL tasks show that V2A delivers significant performance improvements over baseline algorithms. The main

limitations of our work include: (1) additional training time is required for the temporally-consistent modality representation learning process; (2) the theoretical analysis relies on some mild assumptions.

## Acknowledgments

This research was supported by the Research Grants Council of Hong Kong, China (GRF 11217925, GRF 16209124) and the National Natural Science Foundation of China (Grant 72371214). The authors would also like to thank the anonymous reviewers for their valuable comments.

## Impact Statement

This paper presents work whose goal is to advance the field of Machine Learning. There are many potential societal consequences of our work, none of which we feel must be specifically highlighted here.

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

## A. Related Work

**Offline RL.** In offline RL, only a static dataset is accessible and no online interactions are allowed. Therefore, typical off-policy RL algorithms could suffer from the extrapolation error and exhibit poor performance (Kumar et al., 2020; Fujimoto et al., 2019). Common solutions for this challenge include incorporating policy constraints (Kumar et al., 2019; Fujimoto & Gu, 2021), learning a conservative value function (Kumar et al., 2020; Lyu et al., 2022; Jin et al., 2021; Zhang et al., 2024), leveraging dynamics models to facilitate policy learning (Yu et al., 2020; 2021; Qiao et al., 2025a; 2024; 2026), performing in-sample learning (Kostrikov et al., 2021; Xu et al., 2023a; Garg et al., 2023; Zhang et al., 2023), etc. There is another line of work using decision transformer (Chen et al., 2021a; Wu et al., 2023; Xu et al., 2025) for offline sequential decision making. However, these methods require that the offline dataset contains a large amount of data. In contrast, we focus on cross-domain offline RL, where the target data is limited.

**Cross-domain RL.** Cross-domain RL (Niu et al., 2024) aims to generalize or transfer policies across varied domains with distinct agent embodiment (Liu et al., 2022b; Zhang et al., 2020), observation space (Gamrian & Goldberg, 2019; Bousmalis et al., 2018; Ge et al., 2023; Zhang et al., 2021; Hansen et al., 2020), viewpoints (Liu et al., 2018; Sadeghi et al., 2018), transition dynamics (Viano et al., 2021; Liu et al., 2022a), and so on. In this work, we focus on cross-domain RL under dynamics shifts. Previous studies address this challenge by modifying the reward function of the source domain (Liu et al., 2022a; Xue et al., 2023; Eysenbach et al., 2020; Wang et al., 2024b; Yan et al., 2026), adaptively penalizing the $Q$-function on source domain samples (Niu et al., 2022; Qiao et al., 2025c), measuring the dynamics mismatch via representation learning (Lyu et al., 2024a), value discrepancy (Xu et al., 2023b), flow matching (Kong et al., 2025), and so on. In this work, we explore the offline setting where the source and target domains are both offline. Existing works address this issue by conducting dynamics-aware data filtering from the perspective of mutual information (Wen et al., 2024), standard optimal transport (Lyu et al., 2025), unbalanced optimal transport (Chen et al., 2026), and augmenting the source dataset with dynamics-aligned data (Guo et al., 2025b; Van et al., 2025), etc. These works solely focus on dynamics alignment, while DVDF (Qiao et al., 2025b) highlights the significance of value alignment to select high-quality source data. In this work, we further investigate cross-domain offline RL with a heterogeneous source dataset, wherein value alignment can be undermined and value misassignment might occur.

**Multi-modality in Offline RL.** Offline RL often faces the challenge of multi-modal behavior policy in the offline dataset, which can lead the learned policy to converge toward a suboptimal mode. To address this challenge, PLAS (Zhou et al., 2021) and LAPO (Chen et al., 2022) learn to decode actions from a VAE latent space. Additionally, Yang et al. (2022) and Wang et al. (2023) leverage GANs (Goodfellow et al., 2020) to model multiple action modes; Wang et al. (2022) and Akimov et al. (2022) utilize strong generative models such as normalizing flows and diffusion models to model multi-modal distributions. Other related studies include DMPO (Osa et al., 2023) that learns a mixture of deterministic policies, TD3+RKL (Cai et al., 2022) that narrows action coverage by exploiting the mode-seeking property of reverse KL-divergence, and LOM (Wang et al., 2024a) that performs weighted imitation learning on a single promising mode, and so on. Our work is orthogonal to these studies, as we focus on the cross-domain offline RL setting. Moreover, we investigate a novel setting where both the behavior policy and the transition dynamics in the source dataset exhibit multi-modal characteristics.

## B. Proofs of Propositions

In this part, we formally present the missing proofs for our theoretical results in the main text. We also list some useful lemmas for our proofs in Appendix C.

### B.1. Proof of Proposition 4.5

We first present Lemma B.1 below, which gives the suboptimality gap on the source domain with unimodal dynamics.

**Lemma B.1** (suboptimality gap on source domain with unimodal dynamics). *Let $\mathcal{M}_{\mathrm{src}}$ and $\mathcal{M}_{\mathrm{tar}}$ denote the MDPs of the source domain and target domain, where $\mathcal{M}_{\mathrm{src}}$ has unimodal dynamics. For a policy $\hat{\pi}$ trained on $\mathcal{M}_{\mathrm{src}}$, the suboptimality gap of $\hat{\pi}$ on $\mathcal{M}_{\mathrm{tar}}$ can be bounded as follows,*

$$\mathrm{SubOpt}(\hat{\pi}) \leq |J_{\mathcal{M}_{\mathrm{src}}}(\hat{\pi}) - J_{\mathcal{M}_{\mathrm{src}}}(\pi^{\star}_{\mathrm{src}})| + C \cdot \sup_{s,a} [D_{\mathrm{TV}}(P_{\mathrm{src}}(\cdot|s,a), P_{\mathrm{tar}}(\cdot|s,a))], \tag{10}$$

*where $\pi^{\star}_{\mathrm{src}}$ is the optimal policy in $\mathcal{M}_{\mathrm{src}}$, and $C$ is a positive constant.*

*Proof.* Equation 10 can be obtained by slight modifications to Proposition 4.1 in Qiao et al. (2025b). Specifically, the

suboptimality gap can be decomposed as follows,

$$
\begin{aligned}
\mathrm{SubOpt}(\hat{\pi}) &:= J_{\mathcal{M}_{\mathrm{tar}}}(\hat{\pi}) - J_{\mathcal{M}_{\mathrm{tar}}}(\pi^{\star}_{\mathrm{tar}}) \\
&\leq \underbrace{|J_{\mathcal{M}_{\mathrm{src}}}(\hat{\pi}) - J_{\mathcal{M}_{\mathrm{src}}}(\pi^{\star}_{\mathrm{src}})|}_{(\mathrm{I})} + \underbrace{|J_{\mathcal{M}_{\mathrm{tar}}}(\hat{\pi}) - J_{\mathcal{M}_{\mathrm{src}}}(\hat{\pi})|}_{(\mathrm{II})} + \underbrace{|J_{\mathcal{M}_{\mathrm{tar}}}(\pi^{\star}_{\mathrm{src}}) - J_{\mathcal{M}_{\mathrm{src}}}(\pi^{\star}_{\mathrm{tar}})|}_{(\mathrm{III})}.
\end{aligned}
$$

According to Lemma C.1, term (II) can be bounded by:

$$
\begin{aligned}
(\mathrm{II}) &:= |J_{\mathcal{M}_{\mathrm{tar}}}(\hat{\pi}) - J_{\mathcal{M}_{\mathrm{src}}}(\hat{\pi})| \\
&\leq \frac{2\gamma r_{\max}}{(1-\gamma)^2} \cdot \sup_{s,a}[D_{\mathrm{TV}}(P_{\mathrm{src}}(\cdot|s,a), P_{\mathrm{tar}}(\cdot|s,a))].
\end{aligned}
$$

According to Lemma C.2, term (III) can be bounded by:

$$
\begin{aligned}
(\mathrm{III}) &:= |J_{\mathcal{M}_{\mathrm{tar}}}(\pi^{\star}_{\mathrm{tar}}) - J_{\mathcal{M}_{\mathrm{src}}}(\pi^{\star}_{\mathrm{src}})| \\
&\leq \frac{2 r_{\max}}{(1-\gamma)^2} \cdot \sup_{s,a}[D_{\mathrm{TV}}(P_{\mathrm{src}}(\cdot|s,a), P_{\mathrm{tar}}(\cdot|s,a))].
\end{aligned}
$$

We can obtain the desired bound by adding terms (II) and (III). Then we conclude the proof. □

Then we restate and prove our Proposition 4.5 below.

**Proposition B.2** (Proposition 4.5). *Denote the MDP of the target domain and multiple source domains as $\mathcal{M}_{\mathrm{tar}}$ and $\{\mathcal{M}^i_{\mathrm{src}}\}_{i=1}^n$. For a policy $\hat{\pi}$ trained on $\mathcal{D}_{\mathrm{src}}$ defined in Definition 4.2, under some mild assumptions, the suboptimality gap on $\mathcal{M}_{\mathrm{tar}}$ can be bounded as*

$$
\mathrm{SubOpt}(\hat{\pi}) \leq \sum_{i=1}^{n} \alpha_i \cdot \left( \underbrace{|J_{\mathcal{M}^i_{\mathrm{src}}}(\mu^i_{\mathrm{src}}) - J_{\mathcal{M}^i_{\mathrm{src}}}(\pi^{i\star}_{\mathrm{insrc}})|}_{\text{value misalignment}} + C \cdot \underbrace{\sup_{s,a} \left[ D_{\mathrm{TV}}(P^i_{\mathrm{src}}(\cdot|s,a), P_{\mathrm{tar}}(\cdot|s,a)) \right]}_{\text{dynamics misalignment}} \right) + \Delta,
$$

*where $\Delta := \sum_{i=1}^{n} \alpha_i \cdot (\epsilon^{i\star}_{\mathrm{insrc}} + \epsilon^i_{\mu_{\mathrm{src}}})$, $\epsilon^{i\star}_{\mathrm{insrc}} := |J_{\mathcal{M}^i_{\mathrm{src}}}(\pi^{\star}_{\mathrm{insrc}}) - J_{\mathcal{M}^i_{\mathrm{src}}}(\pi^{\star}_{\mathrm{src}})|$, $\epsilon^i_{\mu_{\mathrm{src}}} := |J_{\mathcal{M}^i_{\mathrm{src}}}(\hat{\pi}) - J_{\mathcal{M}^i_{\mathrm{src}}}(\mu^i_{\mathrm{src}})|$, $\pi^{i\star}_{\mathrm{insrc}}$ is the in-sample optimal policy on $\mathcal{D}^i_{\mathrm{src}}$, $\pi^{i\star}_{\mathrm{src}}$ is the optimal policy in $\mathcal{M}^i_{\mathrm{src}}$, $C$ is a constant, and $\{\alpha_i\}_{i=1}^n$ are the mixing coefficients for sub-datasets.*

*Proof.* We first extend the results of Lemma B.1 to the heterogeneous setting. Specifically, denote the MDP of the mixed source domain and each component as $\mathcal{M}_{\mathrm{src}} = (\mathcal{S}, \mathcal{A}, P_{\mathrm{src}}, r, \rho, \gamma)$ and $\{\mathcal{M}^i_{\mathrm{src}} = (\mathcal{S}, \mathcal{A}, P^i_{\mathrm{src}}, r, \rho, \gamma)\}_{i=1}^n$, respectively, where

$$
P_{\mathrm{src}}(\cdot|s,a) = \sum_{i=1}^{n} \alpha_i \cdot P^i_{\mathrm{src}}(\cdot|s,a).
$$

Then according to Lemma B.1, the suboptimality gap can be bounded by:

$$
\mathrm{SubOpt}(\hat{\pi}) \leq \underbrace{|J_{\mathcal{M}_{\mathrm{src}}}(\hat{\pi}) - J_{\mathcal{M}_{\mathrm{src}}}(\pi^{\star}_{\mathrm{src}})|}_{(\mathrm{I})} + C \cdot \underbrace{\sup_{s,a}[D_{\mathrm{TV}}(P_{\mathrm{src}}(\cdot|s,a), P_{\mathrm{tar}}(\cdot|s,a))]}_{(\mathrm{II})}. \tag{11}
$$

We first analyze term (II),

$$
\begin{aligned}
D_{\text{TV}}(P_{\text{src}}(\cdot|s,a), P_{\text{tar}}(\cdot|s,a)) &= D_{\text{TV}}\left(\sum_{i=1}^{n} \alpha_i \cdot P_{\text{src}}^i(\cdot|s,a), P_{\text{tar}}(\cdot|s,a)\right) \\
&= \frac{1}{2}\int_{s'}\left|\sum_{i=1}^{n}\alpha_i \cdot P_{\text{src}}^i(\cdot|s,a) - P_{\text{tar}}(\cdot|s,a)\right| \\
&= \frac{1}{2}\int_{s'}\left|\sum_{i=1}^{n}\alpha_i \cdot \left(P_{\text{src}}^i(\cdot|s,a) - P_{\text{tar}}(\cdot|s,a)\right)\right| \\
&\leq \frac{1}{2}\sum_{i=1}^{n}\alpha_i \int_{s'}\left|P_{\text{src}}^i(\cdot|s,a) - P_{\text{tar}}(\cdot|s,a)\right| \\
&= \sum_{i=1}^{n}\alpha_i \cdot D_{\text{TV}}(P_{\text{src}}^i(\cdot|s,a), P_{\text{tar}}(\cdot|s,a)).
\end{aligned}
\tag{12}
$$

Therefore, term (II) can be further bounded as

$$
\begin{aligned}
\text{(II)} := \sup_{s,a}&[D_{\text{TV}}(P_{\text{src}}(\cdot|s,a), P_{\text{tar}}(\cdot|s,a))] \\
&\leq \sup_{s,a}\left[\sum_{i=1}^{n}\alpha_i \cdot D_{\text{TV}}(P_{\text{src}}^i(\cdot|s,a), P_{\text{tar}}(\cdot|s,a))\right] \\
&\leq \sum_{i=1}^{n}\alpha_i \cdot \sup_{s,a}[D_{\text{TV}}(P_{\text{src}}^i(\cdot|s,a), P_{\text{tar}}(\cdot|s,a))].
\end{aligned}
$$

We then analyze term (I). Since $J_{\mathcal{M}}(\hat{\pi}) = \mathbb{E}_{s\sim\rho}[V_{\mathcal{M}}^{\hat{\pi}}(s)]$, we can focus on the value function:

$$
V_{\mathcal{M}_{\text{src}}}^{\star} - V_{\mathcal{M}_{\text{src}}}^{\hat{\pi}} = \sum_{i=1}^{n}\alpha_i \cdot \left(V_{\mathcal{M}_{\text{src}}^i}^{\star} - V_{\mathcal{M}_{\text{src}}^i}^{\hat{\pi}}\right) + \underbrace{\left(V_{\mathcal{M}_{\text{src}}}^{\star} - V_{\mathcal{M}_{\text{src}}}^{\hat{\pi}}\right) - \sum_{i=1}^{n}\alpha_i \cdot \left(V_{\mathcal{M}_{\text{src}}^i}^{\star} - V_{\mathcal{M}_{\text{src}}^i}^{\hat{\pi}}\right)}_{\text{(III)}}.
$$

We would like to analyze the sign of term (III). To achieve this, we define a function

$$
F(P) := (V^{\star} - V^{\hat{\pi}})(P) = V_P^{\star} - V_P^{\hat{\pi}}.
$$

Then term (III) is equivalent to

$$
\begin{aligned}
\text{(III)} &= F(P_{\text{src}}) - \sum_{i=1}^{n}\alpha_i \cdot F(P_{\text{src}}^i) \\
&= F\left(\sum_{i=1}^{n}\alpha_i \cdot P_{\text{src}}^i\right) - \sum_{i=1}^{n}\alpha_i \cdot F(P_{\text{src}}^i).
\end{aligned}
$$

It is easy to find that $F(P) \geq 0$ since the optimal value $V^{\star}$ is no smaller than $V^{\hat{\pi}}$, and the equality holds if and only if $P = P_0$ where $\hat{\pi}$ is the optimal policy with respect to dynamics $P_0$. Thus, $P_0$ is a global minimizer. By the first-order necessary condition of the minimizer, we have $\nabla F(P_0) = 0$. Furthermore, we assume $F(P)$ satisfies the second-order sufficient condition (SOSC) at the minimizer $P_0$, that is, the Hessian at $P_0$ is positive definite. We note that SOSC is a common condition in numerical optimization (Nocedal & Wright, 2006). Letting $\lambda_{\min}(\cdot)$ denote the minimum eigenvalue operator, we have $\lambda_{\min}(\nabla^2 F(P_0)) = \tilde{\lambda} > 0$ given the positive definite condition. By Weyl's inequality (Franklin, 2000), the minimum eigenvalue operator is continuous. Therefore, for any $\epsilon \in (0, \tilde{\lambda})$, there exists a $\delta > 0$, such that for all $P$ with $\|P - P_0\| < \delta$:

$$
\left|\lambda_{\min}(\nabla^2 F(P)) - \tilde{\lambda}\right| < \epsilon.
$$

By choosing $\epsilon = \tilde{\lambda}/2$, we have $\lambda_{\min}(\nabla^2 F(P)) > \tilde{\lambda}/2 > 0$ for all $P$ in the $\delta$-neighborhood. Thus $F(P)$ is locally convex in the $\delta$-neighborhood of $P_0$. Since $\hat{\pi}$ is trained on $P_{\text{src}}$ and the source domain datasets contain sufficient data, we make a

reasonable assumption that $P_0 \approx P_{\text{src}}$, $\{P^i_{\text{src}}\}^n_{i=1}$, and $P_{\text{src}}$ are all in this neighborhood. By Jensen's inequality (Boyd & Vandenberghe, 2004), we have

$$\text{(III)} = F\left(\sum_{i=1}^n \alpha_i \cdot P^i_{\text{src}}\right) - \sum_{i=1}^n \alpha_i \cdot F(P^i_{\text{src}}) < 0.$$

Therefore, we have

$$V^\star_{\mathcal{M}_{\text{src}}} - V^{\hat{\pi}}_{\mathcal{M}_{\text{src}}} < \sum_{i=1}^n \alpha_i \cdot \left(V^\star_{\mathcal{M}^i_{\text{src}}} - V^{\hat{\pi}}_{\mathcal{M}^i_{\text{src}}}\right).$$

That is,

$$|J_{\mathcal{M}_{\text{src}}}(\hat{\pi}) - J_{\mathcal{M}_{\text{src}}}(\pi^\star_{\text{src}})| \le \sum_{i=1}^n \alpha_i \cdot |J_{\mathcal{M}^i_{\text{src}}}(\hat{\pi}) - J_{\mathcal{M}^i_{\text{src}}}(\pi^{i\star}_{\text{src}})|. \tag{13}$$

Let $\epsilon^{i\star}_{\text{insrc}} := |J_{\mathcal{M}^i_{\text{src}}}(\pi^\star_{\text{insrc}}) - J_{\mathcal{M}^i_{\text{src}}}(\pi^\star_{\text{src}})|$ denote the inherent performance difference between the optimal policy in $\mathcal{M}^i_{\text{src}}$ and the in-sample optimal policy on $\mathcal{D}^i_{\text{src}}$, and $\epsilon^i_{\mu_{\text{src}}} := |J_{\mathcal{M}^i_{\text{src}}}(\hat{\pi}) - J_{\mathcal{M}^i_{\text{src}}}(\mu^i_{\text{src}})|$ denote the performance difference between the learned policy and the behavior policy, then Equation 13 could be further bounded by:

$$|J_{\mathcal{M}_{\text{src}}}(\hat{\pi}) - J_{\mathcal{M}_{\text{src}}}(\pi^\star_{\text{src}})| \le \sum_{i=1}^n \alpha_i \cdot |J_{\mathcal{M}^i_{\text{src}}}(\mu^i_{\text{src}}) - J_{\mathcal{M}^i_{\text{src}}}(\pi^{i\star}_{\text{insrc}})| + \underbrace{\sum_{i=1}^n \alpha_i \cdot (\epsilon^{i\star}_{\text{insrc}} + \epsilon^i_{\mu_{\text{src}}})}_{:=\Delta}. \tag{14}$$

Note that for offline RL algorithms such as IQL (Kostrikov et al., 2021), we typically require the learned policy $\hat{\pi}$ to be close to the behavior policy $\mu_{\text{src}}$, i.e., $D_{\text{KL}}(\hat{\pi}, \mu_{\text{src}}) < \epsilon$. Since

$$D_{\text{KL}}(\hat{\pi}, \mu_{\text{src}}) = D_{\text{KL}}\left(\hat{\pi}, \sum_{i=1}^n \alpha_i \cdot \mu^i_{\text{src}}\right) \le \sum_{i=1}^n \alpha_i \cdot D_{\text{KL}}(\hat{\pi}, \mu^i_{\text{src}}),$$

we make a mild assumption that $D_{\text{KL}}(\hat{\pi}, \mu^i_{\text{src}}) < \epsilon$ for $i = 1, 2, ..., n$, which is a sufficient condition for the requirement $D_{\text{KL}}(\hat{\pi}, \mu_{\text{src}}) < \epsilon$. Under such an assumption, following a similar proof procedure in Proposition 3 in Lyu et al. (2022) and the performance difference lemma (Kakade & Langford, 2002), $\epsilon^i_{\mu_{\text{src}}}$ could be bounded by

$$\epsilon^i_{\mu_{\text{src}}} := \left|J_{\mathcal{M}^i_{\text{src}}}(\hat{\pi}) - J_{\mathcal{M}^i_{\text{src}}}(\mu^i_{\text{src}})\right| \le \mathcal{O}\left(\frac{1}{(1-\gamma)^2}\right).$$

In the meantime, $\epsilon^{i\star}_{\text{insrc}}$ could be seen as a constant for a given sub-dataset $\mathcal{D}^i_{\text{src}}$. Therefore, $\Delta$ is a bounded term. Substituting Equation 12 and Equation 14 into Equation 11, we can get the desired bound and conclude the proof. □

### B.2. Proof of Proposition 5.1

*Proof.* Let $\tau = (s_0, a_0, s_1, ..., s_T)$ be a trajectory sampled from the source dataset $\mathcal{D}_{\text{src}}$. We aim to maximize the log-likelihood $\log p_\theta(\tau)$. By using the law of total probability, we express the log-likelihood as

$$\log p_\theta(\tau) = \log \int p_\theta(\tau, z) dz,$$

where $z$ is the latent variable. Multiplying and Dividing the integrand by the variational posterior $q_\psi(z|\tau)$, we obtain

$$\log p_\theta(\tau) = \log \int q_\psi(z|\tau) \frac{p_\theta(\tau, z)}{q_\psi(z|\tau)} dz = \log \mathbb{E}_{z \sim q_\psi(\cdot|\tau)}\left[\frac{p_\theta(\tau, z)}{q_\psi(z|\tau)}\right].$$

Using Jensen's inequality (Boyd & Vandenberghe, 2004), we obtain the lower bound of $\log p_\theta(\tau)$ as

$$\log p_\theta(\tau) \ge \mathbb{E}_{z \sim q_\psi(\cdot|\tau)}\left[\log \frac{p_\theta(\tau, z)}{q_\psi(z|\tau)}\right].$$

Using the definition of conditional probability, we have $p_\theta(\tau, z) = p_\theta(\tau|z)p(z)$. Therefore, the right-hand side can be formulated as

$$\mathbb{E}_{z \sim q_\psi(\cdot|\tau)} \left[ \log \frac{p_\theta(\tau, z)}{q_\psi(z|\tau)} \right] = \mathbb{E}_{z \sim q_\psi(\cdot|\tau)} \left[ \log \frac{p_\theta(\tau|z)p(z)}{q_\psi(z|\tau)} \right]$$

$$= \mathbb{E}_{z \sim q_\psi(\cdot|\tau)} [\log p_\theta(\tau|z)] - \mathbb{E}_{z \sim q_\psi(\cdot|\tau)} \left[ \log \frac{q_\psi(z|\tau)}{p(z)} \right]$$

$$= \mathbb{E}_{z \sim q_\psi(\cdot|\tau)} [\log p_\theta(\tau|z)] - D_{\mathrm{KL}}(q_\psi(\cdot|\tau), p(\cdot)),$$

which futher leads to

$$\log p_\theta(\tau) \geq \mathbb{E}_{z \sim q_\psi(\cdot|\tau)} [\log p_\theta(\tau|z)] - D_{\mathrm{KL}}(q_\psi(\cdot|\tau), p(\cdot)).$$

Assuming the Markov property for the transition dynamics given the latent representation $z$, we have

$$p_\theta(\tau|z) = p(s_0) \prod_{t=0}^{T-1} p_\theta(s_{t+1}|s_t, a_t, z).$$

Taking the logarithm of both sides and omitting the initial state distribution $p(s_0)$ (which is constant w.r.t. $\theta$ and $z$), we have

$$\log p_\theta(\tau|z) = \sum_{t=0}^{T-1} \log p_\theta(s_{t+1}|s_t, a_t, z).$$

Then the temporally-consistent ELBO for each trajectory $\tau$ can be expressed as

$$\mathsf{TC\text{-}ELBO}_{\psi,\theta}(\tau) := \sum_{t=0}^{T-1} \mathbb{E}_{z \sim q_\psi(\cdot|\tau)} [\log p_\theta(s_{t+1}|s_t, a_t, z)] - D_{\mathrm{KL}}(q_\psi(\cdot|\tau), p(\cdot)).$$

Taking the expectation over all trajectories $\tau \in \mathcal{D}_{\mathrm{src}}$ yields our temporally-consistent ELBO as follows:

$$\mathsf{TC\text{-}ELBO}_{\psi,\theta} := \mathbb{E}_{\tau \sim \mathcal{D}_{\mathrm{src}}}[\mathsf{TC\text{-}ELBO}_{\psi,\theta}(\tau)]$$

$$= \mathbb{E}_{\tau \sim \mathcal{D}_{\mathrm{src}}} \left[ \sum_{t=0}^{T-1} \mathbb{E}_{z \sim q_\psi(\cdot|\tau)} [\log p_\theta(s_{t+1}|s_t, a_t, z)] - D_{\mathrm{KL}}(q_\psi(\cdot|\tau), p(\cdot)) \right]$$

$$= \mathbb{E}_{\tau \sim \mathcal{D}_{\mathrm{src}}, z \sim q_\psi(\cdot|\tau)} \left[ \sum_{t=0}^{T-1} \log p_\theta(s_{t+1} \mid s_t, a_t, z) - D_{\mathrm{KL}}(q_\psi(\cdot \mid \tau), p(\cdot)) \right].$$

This completes our proof. □

## C. Useful Lemmas

**Lemma C.1.** *Denote $\mathcal{M}_1 = (\mathcal{S}, \mathcal{A}, P_1, r, \rho, \gamma)$ and $\mathcal{M}_2 = (\mathcal{S}, \mathcal{A}, P_2, r, \rho, \gamma)$ as two MDPs that only differ in transition dynamics. Then for any policy $\pi$, we have*

$$|J_{\mathcal{M}_1}(\pi) - J_{\mathcal{M}_2}(\pi)| \leq C \cdot \sup_{s,a}[D_{\mathrm{TV}}(P_1(\cdot|s, a), P_2(\cdot|s, a))],$$

*where $C = \frac{2\gamma r_{\max}}{(1-\gamma)^2}$ is a positive constant.*

*Proof.* Please see the proof of Lemma 4.1 in Qiao et al. (2025b). □

**Lemma C.2.** *Denote $\mathcal{M}_1 = (\mathcal{S}, \mathcal{A}, P_1, r, \rho, \gamma)$ and $\mathcal{M}_2 = (\mathcal{S}, \mathcal{A}, P_2, r, \rho, \gamma)$ as two MDPs that only differ in transition dynamics. We also denote the optimal policies of MDPs $\mathcal{M}_1$ and $\mathcal{M}_2$ as $\pi_1^\star$ and $\pi_2^\star$. Then we have*

$$|J_{\mathcal{M}_1}(\pi_1^\star) - J_{\mathcal{M}_2}(\pi_2^\star)| \leq C_2 \cdot \sup_{s,a}[D_{\mathrm{TV}}(P_1(\cdot|s, a), P_2(\cdot|s, a))],$$

*where $C_2 = \frac{2r_{\max}}{(1-\gamma)^2}$ is a positive constant.*

*Proof.* The proof mainly follows that of Proposition 4.1 in Qiao et al. (2025b). Since $J_{\mathcal{M}}(\pi) = \mathbb{E}_{s\sim\rho}[V_{\mathcal{M}}^\pi(s)]$, we have $|J_{\mathcal{M}_1}(\pi_1^\star) - J_{\mathcal{M}_2}(\pi_2^\star)| = |\mathbb{E}_\rho(V_{\mathcal{M}_1}^\star(s) - V_{\mathcal{M}_2}^\star(s))|$. According to Equation (21) in Qiao et al. (2025b), we have

$$\max_{s\in\mathcal{S}}\left|V_{\mathcal{M}_1}^\star(s) - V_{\mathcal{M}_2}^\star(s)\right| \leq \frac{2r_{\max}}{(1-\gamma)^2}\sup_{s,a}[D_{\mathrm{TV}}(P_1(\cdot|s,a), P_2(\cdot|s,a))].$$

Therefore, we have

$$\begin{aligned}
|J_{\mathcal{M}_1}(\pi_1^\star) - J_{\mathcal{M}_2}(\pi_2^\star)| &= |\mathbb{E}_\rho(V_{\mathcal{M}_1}^\star(s) - V_{\mathcal{M}_2}^\star(s))| \\
&\leq \max_{s\in\mathcal{S}}\left|V_{\mathcal{M}_1}^\star(s) - V_{\mathcal{M}_2}^\star(s)\right| \\
&\leq \frac{2r_{\max}}{(1-\gamma)^2}\sup_{s,a}[D_{\mathrm{TV}}(P_1(\cdot|s,a), P_2(\cdot|s,a))].
\end{aligned}$$

Then we conclude the proof. □

## D. Environmental Setting

### D.1. Tasks and Datasets

**Target domain and datasets.** We adopt four locomotion tasks from MuJoCo (Todorov et al., 2012) as the target domain tasks: `halfcheetah-v2`, `hopper-v2`, `walker2d-v2` and `ant-v3`. Since cross-domain offline RL allows only a small quantity of target domain data, we sample 10% data from D4RL (Fu et al., 2020) datasets as the target domain datasets. We allow the target domain datasets to contain data with three data qualities for each task: the **medium** datasets that contain samples collected by an early-stopped SAC (Haarnoja et al., 2018) policy; the **expert** datasets that are collected by an SAC policy trained to the expert level; and the **medium-expert** datasets that mix the medium data and expert data at a 50-50 ratio.

**Source domain and datasets.** We consider two forms of dynamics shifts across four MuJoCo environments (`halfcheetah-v2`, `hopper-v2`, `walker2d-v2`, and `ant-v3`): kinematic and morphology shifts. Kinematic shifts simulate a situation where certain joints are frozen and cannot move, effectively representing a broken robot. Morphology shifts, on the other hand, involve changing the physical structure of the robot compared to the target domain. We further introduce two levels of shifts for each shift type: `easy` and `medium`, based on the magnitude of the shift. Thus, there are four types of dynamics shifts in total: `kinematic-easy`, `kinematic-medium`, `morphology-easy`, and `morphology-medium`. Details regarding the code modifications for these dynamics shifts are provided in the following section.

Regarding the source domain datasets, we adopt a heterogeneous cross-domain offline setting. Specifically, the data in the source domain datasets is collected from distinct source domains using diverse behavior policies. To simulate this setting, for each task, we collect data of two mixed qualities: `medium-replay` and `medium-expert`, across four types of source domains. We then construct the source domain datasets by mixing the data of the same data quality from these four domains in equal proportions (25% each). Consequently, each source domain dataset encapsulates multi-modal dynamics (`kinematic-easy`, `kinematic-medium`, `morphology-easy`, `morphology-medium`) as well as mixed behavior policies (`medium-replay` or `medium-expert`). Each source domain dataset comprises approximately 1M samples, much larger than that of the target domain dataset.

### D.2. Kinematic Shifts Realization

We realize the kinematic shifts by modifying the `.xml` files of the original environments. Specifically, we change the rotation angle of some joints of the robots as follows:

***halfcheetah-kinematic-easy:*** The rotation angle of the joint on the thigh of the robot's back leg is modified from $[-0.52, 1.05]$ to $[-0.052, 0.105]$.

```
# broken back thigh joint
<joint axis="0 1 0" damping="6" name="bthigh" pos="0 0 0" range="-.052 .105"
    stiffness="240" type="hinge"/>
```

***halfcheetah-kinematic-medium:*** The rotation angle of the joint on the thigh of the robot's back leg is modified from $[-0.52, 1.05]$ to $[-0.0052, 0.0105]$.

```
# broken back thigh joint
<joint axis="0 1 0" damping="6" name="bthigh" pos="0 0 0" range="-.0052 .0105"
    stiffness="240" type="hinge"/>
```

***hopper-kinematic-easy***: The rotation angle of the head joint is modified from $[-150, 0]$ to $[-15, 0]$, and the rotation angle of the foot joint is reduced from $[-45, 45]$ to $[-30, -30]$.

```
# broken head joint
<joint axis="0 -1 0" name="thigh_joint" pos="0 0 1.05" range="-15 0" type="hinge"
    />
# broken foot joint
<joint axis="0 -1 0" name="foot_joint" pos="0 0 0.1" range="-30 30" type="hinge"
    />
```

***hopper-kinematic-medium***: The rotation angle of the head joint is modified from $[-150, 0]$ to $[-1.5, 0]$, and the rotation angle of the foot joint is narrowed from $[-45, 45]$ to $[-10, -10]$.

```
# broken head joint
<joint axis="0 -1 0" name="thigh_joint" pos="0 0 1.05" range="-1.5 0" type="hinge
    "/>
# broken foot joint
<joint axis="0 -1 0" name="foot_joint" pos="0 0 0.1" range="-10 10" type="hinge"
    />
```

***walker2d-kinematic-easy***: The rotation angle of the right foot joint is narrowed from $[-45, 45]$ to $[-4.5, 4.5]$.

```
# broken right foot joint
<joint axis="0 -1 0" name="foot_joint" pos="0 0 0.1" range="-4.5 4.5" type="hinge
    "/>
```

***walker2d-kinematic-medium***: The rotation angle of the right foot joint is narrowed from $[-45, 45]$ to $[-0.45, 0.45]$.

```
# broken right foot joint
<joint axis="0 -1 0" name="foot_joint" pos="0 0 0.1" range="-0.45 0.45" type="
    hinge"/>
```

***ant-kinematic-easy***: The rotation angles of the joints on the hip of two front legs are modified from $[-30, 30]$ to $[-3, 3]$.

```
# broken hip joints of front legs
<joint axis="0 0 1" name="hip_1" pos="0.0 0.0 0.0" range="-3 3" type="hinge"/>
<joint axis="0 0 1" name="hip_2" pos="0.0 0.0 0.0" range="-3 3" type="hinge"/>
```

***ant-kinematic-medium***: The rotation angles of the joints on the hip of two front legs are modified from $[-30, 30]$ to $[-0.3, 0.3]$.

```
# broken hip joints of front legs
<joint axis="0 0 1" name="hip_1" pos="0.0 0.0 0.0" range="-0.3 0.3" type="hinge"
    />
<joint axis="0 0 1" name="hip_2" pos="0.0 0.0 0.0" range="-0.3 0.3" type="hinge"
    />
```

### D.3. Morphology Shifts Realization

We change the morphology of the robots by modifying the `.xml` files as follows.

***halfcheetah-morph-easy***: The torso of the robot is set to 0.8 times that in the target domain.

```
# torso
<geom fromto="-.4 0 0 .4 0 0" name="torso" size="0.046" type="capsule"/>
<geom axisangle="0 1 0 .87" name="head" pos=".5 0 .1" size="0.046 .15" type="
    capsule"/>
<body name="bthigh" pos="-.4 0 0">
<body name="fthigh" pos=".4 0 0">
```

*halfcheetah-morph-medium*: The torso of the robot is set to 0.5 times that in the target domain.

```
# torso
<geom fromto="-.4 0 0 .4 0 0" name="torso" size="0.046" type="capsule"/>
<geom axisangle="0 1 0 .87" name="head" pos=".35 0 .1" size="0.046 .15" type="
    capsule"/>
<body name="bthigh" pos="-.25 0 0">
<body name="fthigh" pos=".25 0 0">
```

*hopper-morph-easy*: The torso is increased to 1.5 times that in the target domain, and the length of the torso becomes 0.48.

```
# torso
<geom friction="0.9" fromto="0 0 1.53 0 0 1.05" name="torso_geom" size="0.075"
    type="capsule"/>
```

*hopper-morph-medium*: The torso is increased to 2.0 times that in the target domain, and the length of the torso becomes 0.64.

```
# torso
<geom friction="0.9" fromto="0 0 1.69 0 0 1.05" name="torso_geom" size="0.1" type
    ="capsule"/>
```

*walker2d-morph-easy*: The torso size is changed to 1.5 times that in the target domain, and the length of the torso becomes 0.48.

```
# torso
<geom friction="0.9" fromto="0 0 1.53 0 0 1.05" name="torso_geom" size="0.075"
    type="capsule"/>
```

*walker2d-morph-medium*: The torso is changed to 2.0 times that in the target domain, and the length of the torso becomes 0.64.

```
# torso
<geom friction="0.9" fromto="0 0 1.69 0 0 1.05" name="torso_geom" size="0.1" type
    ="capsule"/>
```

*ant-morph-easy*: The leg sizes of the front two legs are changed to 0.8 times those in the source domain.

```
# leg size
<geom fromto="0.0 0.0 0.0 0.32 0.32 0.0" name="left_ankle_geom" size="0.08" type=
    "capsule"/>
<geom fromto="0.0 0.0 0.0 -0.32 0.32 0.0" name="right_ankle_geom" size="0.08"
    type="capsule"/>
```

*ant-morph-medium*: The sizes of the front two legs are changed to be 0.5 times those in the source domain.

```
# leg size
```

```
<geom fromto="0.0 0.0 0.0 0.2 0.2 0.0" name="left_ankle_geom" size="0.08" type="
    capsule"/>
<geom fromto="0.0 0.0 0.0 -0.2 0.2 0.0" name="right_ankle_geom" size="0.08" type=
    "capsule"/>
```

# E. Implementation Details

In this section, we provide the implementation details of the baselines and our method V2A.

### E.1. Baselines

**IQL:** we train IQL (Kostrikov et al., 2021) on both target and source domain datasets. The state value function is updated by expectile regression:

$$\mathcal{L}_V = \mathbb{E}_{(s,a) \sim \mathcal{D}_{\mathrm{src}} \cup \mathcal{D}_{\mathrm{tar}}} \left[ L_2^\tau (Q_\theta(s,a) - V_\psi(s)) \right], \tag{15}$$

where $L_2^\tau(u) = |\tau - \mathbb{I}(u < 0)||u^2|$, $\mathbb{I}(\cdot)$ is the indicator function. This enables learning an in-sample optimal value function. Then the state-action value function is updated by:

$$\mathcal{L}_Q = \mathbb{E}_{(s,a,r,s') \sim \mathcal{D}_{\mathrm{src}} \cup \mathcal{D}_{\mathrm{tar}}} \left[ (r(s,a) + \gamma V_\psi(s') - Q_\theta(s,a))^2 \right]. \tag{16}$$

Then the advantage function is extracted by $A(s,a) = Q(s,a) - V(s)$. The policy is optimized via exponential advantage weighted behavior cloning:

$$\mathcal{L}_\pi = -\mathbb{E}_{(s,a) \sim \mathcal{D}_{\mathrm{src}} \cup \mathcal{D}_{\mathrm{tar}}} [\exp(\beta \cdot A(s,a)) \log \pi_\phi(a|s)], \tag{17}$$

where $\beta$ is the temperature coefficient. We implement IQL based on its official codebase[1].

**BOSA:** BOSA (Liu et al., 2024) tackles the state-action OOD problem and dynamics OOD problem in cross-domain offline RL with supported policy optimization and supported value optimization, respectively. Specifically, BOSA updates the critic with supported value optimization:

$$\mathcal{L}_Q = \mathbb{E}_{(s,a) \sim \mathcal{D}_{\mathrm{src}}} [Q_\theta(s,a)] + \mathbb{E}_{\substack{(s,a,r,s') \sim \mathcal{D}_{\mathrm{src}} \cup \mathcal{D}_{\mathrm{tar}}, \\ a' \sim \pi_\phi(s')}} \left[ \mathbb{I}(\hat{P}_{\mathrm{tar}}(s'|s,a) > \epsilon)(Q_\theta(s,a) - y)^2 \right],$$

where $\mathbb{I}(\cdot)$ is the indicator function, and $\hat{P}_{\mathrm{tar}}(s'|s,a)$ is the estimated target dynamics. The policy is updated by supported policy optimization to mitigate the OOD action issue:

$$\mathcal{L}_\pi = \mathbb{E}_{s \sim \mathcal{D}_{\mathrm{src}} \cup \mathcal{D}_{\mathrm{tar}}, \, a \sim \pi_\phi(s)} [Q_\theta(s,a)], \quad \text{s.t.} \ \mathbb{E}_{s \sim \mathcal{D}_{\mathrm{src}} \cup \mathcal{D}_{\mathrm{tar}}} [\hat{\pi}_{\mathrm{mix}}(\pi_\phi(s) \mid s)] > \epsilon',$$

where $\hat{\pi}_{\mathrm{mix}}(\cdot|s)$ is the behavior policy of the mixed datasets $\mathcal{D}_{\mathrm{src}} \cup \mathcal{D}_{\mathrm{tar}}$ learned with CVAE (Sohn et al., 2015). We implement BOSA from ODRL[2] benchmark (Lyu et al., 2024b), which provides implementations for various off-dynamics RL algorithms.

**DARA:** DARA (Liu et al., 2022a) employs dynamics-aware reward modification to achieve dynamics adaptation. Specifically, DARA trains two domain classifiers $q_{\theta_{SAS}}(\text{target}|s_t, a_t, s_{t+1})$ and $q_{\theta_{SA}}(\text{target}|s_t, a_t)$ as follows:

$$\mathcal{L}_{\theta_{SAS}} = \mathbb{E}_{\mathcal{D}_{\mathrm{tar}}} [\log q_{\theta_{SAS}}(\text{target}|s_t, a_t, s_{t+1})] + \mathbb{E}_{\mathcal{D}_{\mathrm{src}}} [\log(1 - q_{\theta_{SAS}}(\text{target}|s_t, a_t, s_{t+1}))]$$
$$\mathcal{L}_{\theta_{SA}} = \mathbb{E}_{\mathcal{D}_{\mathrm{tar}}} [\log q_{\theta_{SA}}(\text{target}|s_t, a_t)] + \mathbb{E}_{\mathcal{D}_{\mathrm{src}}} [\log(1 - q_{\theta_{SA}}(\text{target}|s_t, a_t))].$$

The domain classifiers are used to measure the dynamics gap $\log \frac{P_{\mathcal{M}_{\mathrm{tar}}}(s_{t+1}|s_t, a_t)}{P_{\mathcal{M}_{\mathrm{src}}}(s_{t+1}|s_t, a_t)}$ between the source domain and the target domain according to Bayes' rule. Then the estimated dynamics gap serves as a penalty to the source domain rewards:

$$\hat{r}_{\mathrm{DARA}} = r - \lambda \times \delta_r, \quad \delta_r(s_t, a_t) = -\log \frac{q_{\theta_{SAS}}(\text{target}|s_t, a_t, s_{t+1}) q_{\theta_{SA}}(\text{source}|s_t, a_t)}{q_{\theta_{SAS}}(\text{source}|s_t, a_t, s_{t+1}) q_{\theta_{SA}}(\text{target}|s_t, a_t)},$$

where $\lambda$ controls the intensity of the reward penalty. We use the DARA implementation from ODRL and follow the hyperparameter setting in the original paper: $\lambda$ is set to 0.1, and the reward penalty is clipped within $[-10, 10]$.

---

[1]https://github.com/ikostrikov/implicit_q_learning.git
[2]https://github.com/OffDynamicsRL/off-dynamics-rl.git

**IGDF:** IGDF (Wen et al., 2024) quantifies the domain discrepancy between the source domain and the target domain with contrastive representation learning. To facilitate effective knowledge transfer, IGDF implements data filtering to selectively share source domain samples exhibiting smaller dynamics gaps. Specifically, IGDF trains a score function $h(\cdot)$ using $(s, a, s'_{\text{tar}}) \sim \mathcal{D}_{\text{tar}}$ as the positive samples, and transitions $(s, a, s'_{\text{src}})$ as the negative samples, where $(s, a) \sim \mathcal{D}_{\text{tar}}$ and $s'_{\text{src}} \sim \mathcal{D}_{\text{src}}$. $h(\cdot)$ is optimized via the following contrastive learning objective:

$$\mathcal{L} = -\mathbb{E}_{(s,a,s'_{\text{tar}})}\mathbb{E}_{s'_{\text{src}}} \left[ \log \frac{h(s, a, s'_{\text{tar}})}{\sum_{s' \in s'_{\text{tar}} \cup s'_{\text{src}}} h(s, a, s')} \right].$$

Based on the learned score function, IGDF proposes to selectively share source domain data for training value functions:

$$\mathcal{L}_Q = \frac{1}{2}\mathbb{E}_{\mathcal{D}_{\text{tar}}} \left[ (Q_\theta - \mathcal{T}Q_\theta)^2 \right] + \frac{1}{2}\alpha \cdot h(s, a, s')\mathbb{E}_{(s,a,s')\sim\mathcal{D}_{\text{src}}} \left[ \mathbb{I}(h(s, a, s') > h_{\xi\%})(Q_\theta - \mathcal{T}Q_\theta)^2 \right],$$

where $\mathbb{I}(\cdot)$ is the indicator function, $\alpha$ is the weighting coefficient, $\xi$ is the data selection ratio. We implement IGDF based on its official codebase[3].

**OTDF:** OTDF (Lyu et al., 2025) estimates the discrepancy between the source domain and target domain by computing the Wasserstein distance (Gabriel & Marco, 2019):

$$\mathcal{W}(u, u') = \min_{\mu \in M} \sum_{t=1}^{|\mathcal{D}_{\text{src}}|} \sum_{t'=1}^{|\mathcal{D}_{\text{tar}}|} C(u_t, u'_{t'}) \cdot \mu_{t,t'}, \tag{18}$$

where $u = s_{\text{src}} \oplus a_{\text{src}} \oplus s'_{\text{src}}$, $u' = s_{\text{tar}} \oplus a_{\text{tar}} \oplus s'_{\text{tar}}$ are the concatenated vectors, $C$ is the cost function and $M$ is the coupling matrices. After solving Equation 18 for the optimal coupling matrix $\mu^\star$, the OTDF computes the distance between a source domain sample and the target domain dataset via

$$d(u_t) = - \sum_{t'=1}^{|\mathcal{D}_{\text{tar}}|} C(u_t, u'_{t'})\mu^\star_{t,t'}, \quad u_t = (s^t_{\text{src}}, a^t_{\text{src}}, (s'_{\text{src}})^t) \sim \mathcal{D}_{\text{src}}.$$

Then the critic is updated by

$$\mathcal{L}_Q = \mathbb{E}_{\mathcal{D}_{\text{tar}}} \left[ (Q_\theta - \mathcal{T}Q_\theta)^2 \right] + \mathbb{E}_{(s,a,s')\sim\mathcal{D}_{\text{src}}} \left[ \exp(\alpha \times d)\mathbb{I}(d > d_{\%})(Q_\theta - \mathcal{T}Q_\theta)^2 \right].$$

Furthermore, OTDF incorporates a policy regularization term that forces the policy to stay close to the support of the target dataset:

$$\widehat{\mathcal{L}_\pi} = \mathcal{L}_\pi - \beta \times \mathbb{E}_{s\sim\mathcal{D}_{\text{src}}\cup\mathcal{D}_{\text{tar}}} \log \pi^b_{\text{tar}}(\pi(\cdot|s)|s),$$

where $\mathcal{L}_\pi$ is the original policy optimization objective and $\beta$ is the weight coefficient, $\pi^b_{\text{tar}}$ is the behavior policy of the target domain dataset learned with a CVAE. We run the official code[4] of OTDF for implementation.

**DVDF:** DVDF (Qiao et al., 2025b) introduces the concept of value alignment and demonstrates that, in addition to dynamics-aligned source samples, high-quality source samples are also crucial for cross-domain offline RL. Building on this insight, DVDF proposes a scoring function that trades off value alignment and dynamics alignment:

$$g(s, a, s') = \lambda \cdot h(s, a, s') + (1 - \lambda) \cdot A(s, a),$$

where $\lambda$ is the trade-off coefficient, $h(\cdot)$ is computed by IGDF or OTDF, $A(s, a)$ is the advantage function pre-trained on the source domain dataset. Then DVDF performs data filtering for training similar to IGDF and OTDF. We implement DVDF ourselves, following the implementation details provided in the original paper.

### E.2. Implementation Details of V2A

In this part, we present more implementation details of V2A.

---

[3]https://github.com/BattleWen/IGDF.git
[4]https://github.com/dmksjfl/OTDF.git

---

**Algorithm 1** Unifying Value Alignment and Assignment (V2A)

---

1: **Require:** Source domain offline dataset $\mathcal{D}_{\mathrm{src}}$, target domain offline dataset $\mathcal{D}_{\mathrm{tar}}$
2: **Initialization:** Policy network $\pi_\eta$, value network $V_\beta$, $Q$ network $Q_\phi$, target $Q$ network $Q_{\phi'}$, representation encoder $q_\psi$, dynamics decoder $p_\theta$, data selection ratio $\xi$, batch size $B$, importance coefficient $\alpha$, trade-off coefficient $\lambda$
3: **// Temporally-consistent modality representation learning**
4: Train $q_\psi$ and $p_\theta$ on $\mathcal{D}_{\mathrm{src}}$ by iteratively optimizing Equation 4 and Equation 5
5: **// Modality-aware advantage learning**
6: Relabel $\mathcal{D}_{\mathrm{src}}$ with the learned representation $z \sim q_\psi$ to obtain $\mathcal{D}'_{\mathrm{src}}$
7: Learning an advantage function $A(s, a, z)$ on $\mathcal{D}'_{\mathrm{src}}$ in SQL manner
8: **// Obtaining the score function**
9: Obtain $h(s, a, s')$ via contrastive representation learning or optimal transport, obtain the score function $f(s, a, s', z) = \lambda \cdot h(s, a, s') + (1 - \lambda) \cdot A(s, a, z)$
10: **// TD learning**
11: **for** each gradient step **do**
12:     Sample $b_{\mathrm{src}} := \{(s, a, r, s', z)\}$ from $\mathcal{D}'_{\mathrm{src}}$
13:     Sample $b_{\mathrm{tar}} := \{(s, a, r, s')\}$ from $\mathcal{D}_{\mathrm{tar}}$
14:     Sample the top-$\xi$ samples from $b_{\mathrm{src}}$ ranked by $f(s_{\mathrm{src}}, a_{\mathrm{src}}, s'_{\mathrm{src}}, z)$
15:     Compute weights $\omega(s, a, s')$ following:
16:       $\omega(s, a, s', z) = \mathbb{I}(f(s, a, s', z) \geq f_{\xi\%})$
17:     **// Optimize the $V_\beta$ function**
18:     Compute loss $\mathcal{L}_V$:
19:       $\mathcal{L}_V = \mathbb{E}_{(s,a)\sim\mathcal{D}'_{\mathrm{src}}\cup\mathcal{D}_{\mathrm{tar}}} \left[ L_2^\tau \left( Q_\phi(s, a) - V_\beta(s) \right) \right]$
20:     Update $V_\beta$ using $\mathcal{L}_V$
21:     **// Optimize the $Q_\phi$ function**
22:     Compute loss $\mathcal{L}_Q$:
23:       $\mathcal{L}_Q = \frac{1}{2} \cdot \mathbb{E}_{(s,a,r,s')\sim\mathcal{D}_{\mathrm{tar}}} \left[ \left( Q_\phi(s, a) - (r + \gamma V_\beta(s')) \right)^2 \right]$
24:       $+ \frac{1}{2} \cdot \mathbb{E}_{(s,a,r,s',z)\sim\mathcal{D}'_{\mathrm{src}}} \left[ \omega(s, a, s', z) f(s, a, s', z) \left( Q_\phi(s, a) - (r + \gamma V_\beta(s')) \right)^2 \right]$
25:     Update $Q_\phi$ using $\mathcal{L}_Q$
26:     **// Update target network**
27:     Update target network parameters: $\phi' \leftarrow (1 - \mu)\phi + \mu\phi'$
28:     **// Policy extraction (AWR)**
29:     Compute advantage $A(s, a) = Q_\phi(s, a) - V_\beta(s)$
30:     Optimize policy network $\pi_\eta$ using advantage-weighted regression (AWR):
31:       $\mathcal{L}_\pi = \mathbb{E}_{(s,a)\sim\mathcal{D}'_{\mathrm{src}}\cup\mathcal{D}_{\mathrm{tar}}} \left[ \exp(\alpha A(s, a)) \log \pi_\eta(a|s) \right]$
32: **end for**

---

**Learning the modality representation.** We model the representation encoder $q_\psi$ as an RNN network which takes a variable-horizon trajectory $\tau$ as input. The final hidden state of the RNN is used as the output representation $z$. The encoder $q_\psi$ consists of one input layer and three hidden layers, with a hidden size of 256. To mitigate the gradient vanishing problem that arises with long-horizon inputs, we employ an LSTM (Hochreiter & Schmidhuber, 1997) as the backbone of $q_\psi$, simply utilizing the nn.LSTM() module in PyTorch (Paszke et al., 2019). For the dynamics decoder $p_\theta$, we construct it as an ensemble dynamics model with ensemble size of $N$, where each ensemble member outputs a Gaussian distribution over the next state: $\left\{ \widehat{P}_{\psi_i} = \mathcal{N}(\mu_{\psi_i}, \Sigma_{\psi_i}) \right\}_{i=1}^N$. Each ensemble model consists of an input layer, an output layer, and three hidden layers with a size of 256. In the E-step, for each trajectory in $\mathcal{D}_{\mathrm{src}}$, we randomly select one ensemble member to compute the likelihood. In the M-step, we perform maximum likelihood estimation (MLE) on all ensemble members.

**Obtaining the score function.** In V2A, the score function $f(\cdot)$ incorporates a dynamics-aware score function $h(\cdot)$ to measure dynamics alignment and a modality-aware advantage function $A(\cdot)$ to measure value alignment. We note that $h(\cdot)$ can be obtained through various methods, which makes V2A an algorithmic framework that can be integrated with multiple algorithms. In this work, we integrate V2A with IGDF and OTDF, respectively, using contrastive learning or optimal transport to obtain $h(\cdot)$. It is important to note that both $h(\cdot)$ and $A(\cdot)$ are used exclusively in the data filtering stage and are not updated during subsequent policy optimization steps.

**Data filtering and policy optimization.** After we obtain the relabeled source dataset $\mathcal{D}'_{\mathrm{src}}$ and the score function $f(\cdot)$, we perform data filtering when learning the value function, and optimize the policy with IQL on $\mathcal{D}'_{\mathrm{src}} \cup \mathcal{D}_{\mathrm{tar}}$. The detailed pseudo-code for V2A is presented in Algorithm 1.

### E.3. Hyperparameter Setup

We list the major hyperparameter setup for V2A and several baselines in our experiments in Table 3.

*Table 3.* Hyperparameter setup for V2A and baseline methods

| Hyperparameter | Value |
|---|---|
| **Shared** | |
| Learning rate | $3 \times 10^{-4}$ |
| Optimizer | Adam (Kingma, 2014) |
| Discount factor | 0.99 |
| Activation function | ReLU (Nair & Hinton, 2010) |
| Source domain batch size | 128 |
| Target domain batch size | 128 |
| **IGDF** | |
| Encoder pretrained steps | 7000 |
| Importance coefficient | 1.0 |
| Data selection ratio $\xi$ | 25% |
| **OTDF** | |
| CVAE training steps | 10000 |
| Number of sampled latent variables $M$ | 10 |
| Cost function | cosine |
| Data Selection ratio $\xi$ | 80% |
| **DVDF** | |
| SQL pre-training steps | $1 \times 10^{6}$ |
| Trade-off coefficient $\lambda$ | 0.7 |
| Data selection ratio $\xi$ | 50% |
| **V2A** | |
| Representation learning steps | $1 \times 10^{6}$ |
| SQL pre-training steps | $1 \times 10^{6}$ |
| Trade-off coefficient $\lambda$ | 0.6 |
| Data selection ratio $\xi$ | 50% |
| Decoder ensemble size $N$ | 3 |

### E.4. Compute Infrastructure

We present the compute infrastructure for all the experiments in Table 4.

*Table 4.* Compute infrastructure

| CPU | GPU | Memory |
|---|---|---|
| AMD Ryzen Threadripper PRO 5975WX | NVIDIA RTX A6000 ($\times 4$) | 503 GB |

## F. Additional Experimental Results

In this section, we present additional experimental results beyond those reported in the main text.

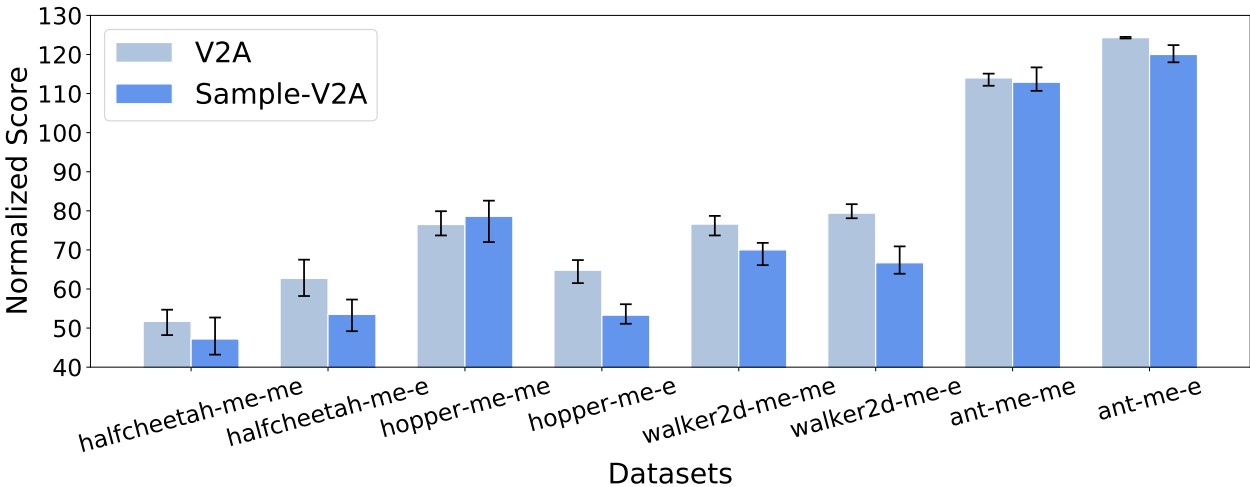

*Figure 4.* Ablation study on the necessity of temporally-consistent ELBO. Each result is averaged over 5 random seeds.

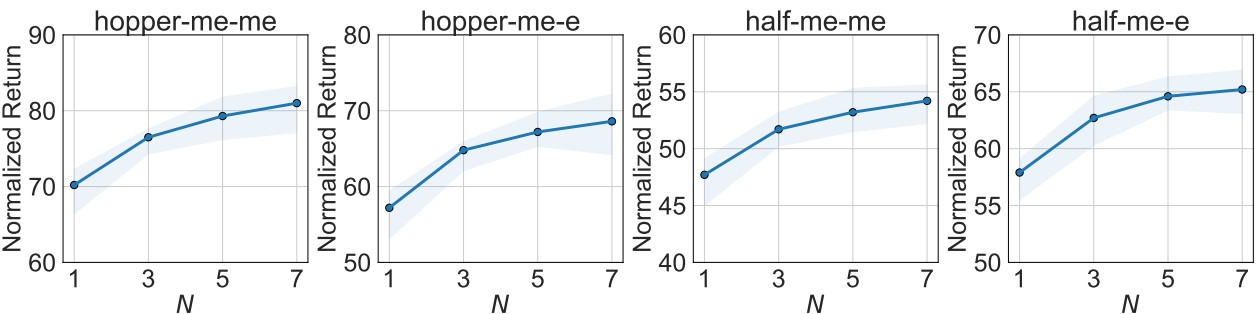

*Figure 5.* Parameter study on the effect of $N$. Results averaged across 5 random seeds.

### F.1. Ablation Study

**Temporally-Consistent ELBO.** We conduct an ablation study on the necessity of temporally-consistent ELBO in Equation 3. Specifically, we model $p_\theta$ as a typical fully-connected neural network and optimize the sample-level ELBO in Equation 2. We then proceed to perform advantage learning using the learned representations, while keeping the rest of the procedure identical to that of V2A. We refer to this variant as sample-V2A. Apart from the algorithmic difference, we keep all other experimental settings the same as those described in Section 6.1, and compare the performance of the original V2A and sample-V2A across multiple datasets. Each experiment is run with 5 random seeds. The experimental results are presented in Figure 4. We observe that the original V2A outperforms sample-V2A in 7 out of the 8 tasks compared. In the `hopper-me-me` task, the original V2A also achieves performance comparable to that of sample-V2A. These results suggest that using the temporally-consistent ELBO yields better performance than the sample-level ELBO.

### F.2. Extended Parameter Study

In Section 6.3, we have investigated the effects of $\lambda$ and $\xi$. In this part, we further examine the impact of the ensemble size $N$ of the dynamics model $p_\theta$.

**Ensemble size $N$.** Ensemble is a commonly used technique to reduce model prediction error (Janner et al., 2019). Typically, a larger ensemble size $N$ leads to higher prediction accuracy for the dynamics model, but it also incurs greater computational overhead during training. We vary $N$ across $\{1, 3, 5, 7\}$ and conduct experiments on four tasks; the results are shown in Figure 5. We observe that V2A achieves lower performance when $N = 1$, which we attribute to the low accuracy of the dynamics model negatively impacting representation learning. Meanwhile, the performance differences among $N = 3, 5, 7$ are not obvious. Therefore, for computational efficiency, we fix $N = 3$ across all tasks.

*Figure 6.* Time cost comparison between V2A and IGDF, OTDF, DVDF.

### F.3. Time Cost Comparison

In Figure 6, we compare the training time required by V2A and baseline methods including IGDF, OTDF, and DVDF across four tasks. We break down the total time into five parts: representation extraction, advantage learning, solving optimal transport (OT) and training the CVAE (for OTDF), contrastive learning (for IGDF), and policy optimization. Each method incorporates a subset of these parts. Compared to other methods, V2A requires a similar amount of time for policy optimization, with the additional overhead mainly concentrated in the representation extraction and advantage learning stages. However, since these steps are decoupled from policy optimization, they can be precomputed, thereby introducing no extra overhead in subsequent experiments.

## G. LLM Usage Declaration

We use LLM to polish the grammar and phrasing of our early draft. We affirm that LLM is not involved in the core contributions of this work, including the development of the method, the proof of theorems, and the experimental parts.

