# OpenReview forum: "Unifying Value Alignment and Assignment in Cross-Domain Offline Reinforcement Learning with Heterogeneous Datasets"
_ICML.cc/2026/Conference — ICML 2026 regular_

### Official Review · Reviewer_WwvE · 2026-02-26

**Soundness:** 3
**Presentation:** 3
**Significance:** 3
**Originality:** 3
**Overall Recommendation:** 4
**Confidence:** 4

**Summary:**

This paper identifies the value misassignment problem in heterogeneous cross-domain offline RL, in which the source may be collected from multiple domains by diverse behavior policies. The value misassignment problem originates from the fact that the value misalignment, which is the suboptimality gap of behavior policies, differs across sub-datasets and is usually not properly captured by simply pretraining on the mixed source dataset. This prevents proper data filtering on multi-domain source datasets. To mitigate the problem, the paper proposes a framework that (i) learns modality representations on the source dataset by maximizing Temporally-Consistent ELBO and (ii) perform modality-aware advantage learning by conditioning on the latent $z$. Empirical results suggest that the proposed framework outperforms baseline methods in the heterogeneous cross-domain setting.

**Compliance With Llm Reviewing Policy:**

Affirmed.

**Final Justification:**

The rebuttal addressed my main concerns.

**Key Questions For Authors:**

1. The parameter change (e.g., the rotation angle scaling factor, torso/leg length scaling factor) for `easy` and `medium` settings seems arbitrary and varies across different tasks. How does one determine that these modifications are "easy/medium"?.
2. Section 3 states that the source domain is created by introducing medium-level kinematic shifts to `halfcheetah-v2`. Does it mean that Figure 1 only shows a single-source domain task? Currently, it is difficult to see this as there are only red and blue colors which denotes different qualities but not different domains. This seems inconsistent with the paper's main narrative as one might expect V2A to work best in tasks with multiple source domains. I am willing to increase my score if the authors can clarify this domain setup and provide additional evidence of V2A's performance in a multi-source context.

**Limitations:**

- The time cost comparison shown in Figure 6 could be briefly mentioned in the main text.

**Strengths And Weaknesses:**

**Strengths**
- The paper highlights the value misassignment problem in heterogeneous cross-domain offline RL. To the best of my knowledge, this concept is novel.
- The paper is technically sound. The proposed method is a reasonable response to the incorrect filtering phenomenon presented in Section 4.2 and Figure 1. Also, the datasets are collected with multiple dynamics, which corresponds the heterogeneous cross-domain setting.
- The paper is clearly written and well organized.

**Weaknesses**
- The connection between the theoretical analysis and the practical algorithm is relatively weak. Proposition 4.5 suggests reweighting sub-datasets with $\alpha_{i}$. However, the proposed algorithm performs filtering on the transition level while conditioning on learned latent $z$.
- The importance of the problem this paper aims to address could be further explained. The experiments in the paper are limited to locomotion tasks in D4RL. It is not clear whether the results would transfer to other scenarios (e.g., navigation, manipulation).
- The criteria for highlighting top-performing results in Table 2 are unclear. For instance, IQL scores 66.3 in `hopper-mr-e` (which is actually higher than IGDF-V2A's 61.7) but is not highlighted. There are other similar cases in which vanilla IQL performs well (e.g., `hopper-me-m`, `ant-me-m`), which raises questions regarding the actual difficulty of these benchmarks.
- It appears that the time cost almost doubles when applying the proposed V2A framework to IGDF/OTDF, as shown in Figure 6.

P.S. On page 6, line 322: "..., we combine them with IGDF and DVDF respectively for performance comparison", please check if DVDF should be OTDF, which is the other standalone algorithm to be combined with in Table 2.

---

> ### Author Rebuttal · Authors · 2026-03-27
>
> We thank Reviewer WwvE for the insightful comments. Below are our responses to the concerns.
>
> **1. On the connection between theoretical analysis and practical algorithm**
>
> We respectfully clarify that Proposition 4.5 is closely connected to our practical algorithm. Specifically, Proposition 4.5 shows that value misassignment leads to an incorrect estimation of the per-source value misalignment term. As a result, directly applying the original DVDF filtering method may assign larger weights ($\alpha_i$) to suboptimal samples, thereby loosening the suboptimality gap. This suggests that accurately estimating the modality-specific value misalignment term is crucial for minimizing the gap. V2A addresses this by introducing modality-aware advantage learning, where the inferred latent dynamics $z$ enables modality-specific value alignment, leading to more appropriate weights over source sub-datasets and thus a smaller suboptimality gap in Proposition 4.5. We will clarify this connection more explicitly in the updated version.
>
> **2. Further explanation on the problem importance**
>
> Our paper is the first to study heterogeneous cross-domain offline RL, which is a novel and general setting, where the transition dynamics and behavior policy in the source datasets can be multi-modal. We further show that directly applying previous cross-domain offline RL methods to this setting is suboptimal, due to the non-trivial value misassignment issue that cannot be addressed by prior works. Our proposed V2A algorithm is specifically designed to address this significant challenge and to maintain effective transfer in the heterogeneous setting. We believe our work addresses a critical problem and provides new insights into cross-domain offline RL.
>
> **3. More evaluation results beyond MuJoCo**
>
> We appreciate this suggestion. We conduct additional experiments on Adroit datasets. Due to space limits, we respectfully refer the reviewer to our response to Reviewer uXVB's Concern 2 for detailed experimental results.
>
>  **4. Unclear highlighting criteria in Table 2**
>
> Thanks for pointing this out. We will correct the highlighting in Table 2 in the updated version, including the cases where IQL achieves good results. We also clarify that the benchmark remains challenging: although IQL performs well on a few tasks, its overall performance remains clearly below V2A. Thus, these cases do not undermine the benchmark's difficulty.
>
> **5. On the time cost of V2A**
>
> We acknowledge that V2A introduces additional computational overhead due to representation learning. However, this cost is mainly incurred in the modality representation learning stage, which is decoupled from policy optimization and thus can be precomputed. Therefore, it does not introduce extra overhead in subsequent experiments. We will clarify this limitation explicitly in the updated version.
>
> **6. Potential typo in Line 322**
>
> Thanks for catching this. It should be OTDF rather than DVDF. We will fix this typo in the updated version.
>
> **7. How to determine easy and medium settings**
>
> The easy/medium settings are determined by the magnitude of the dynamics shift relative to the target domain: the medium setting introduces a larger deviation in rotation angle and torso/leg length than the easy setting, leading to a more challenging transfer scenario. Additionally, the scaling factors follow the ODRL benchmark [1] rather than being chosen arbitrarily, and we will clarify this in the updated version.
>
> **8. On the heterogeneity of the motivation example**
>
> Our source dataset is constructed by mixing (1) 0.5M expert data from the original halfcheetah-v2 domain with a small (zero) dynamics shift and (2) 0.5M medium-quality data from the halfcheetah-kinematic-medium domain with a medium-level dynamics shift. Therefore, it is **already heterogeneous**, containing both multiple transition dynamics  (with 0 & medium shifts) and data qualities (expert & medium data). In Figure 1, the red points denote medium data from the halfcheetah-kinematic-medium domain, while the blue points denote expert data from the original halfcheetah-v2 domain. Thus, Figure 1 already reflects both varying dynamics and data quality information.
>
> To further address the reviewer's concern, we conduct an additional experiment in a clearer heterogeneous setting, where the source dataset mixes 0.5M expert data from halfcheetah-kinematic-easy and 0.5M medium-quality data from halfcheetah-kinematic-medium. Using this heterogeneous source dataset, we train V2A and DVDF and evaluate them on the target domain across 5 seeds, as shown in the table below. The results show that V2A still outperforms DVDF under this multi-source setting. We believe this provides evidence for the effectiveness of V2A in heterogeneous settings.
>
> | | V2A | DVDF |
> |-|-|-|
> |Score| **60.8 $\pm$ 2.1** | 42.6 $\pm$ 3.4 |
>
> [1] ODRL: A Benchmark for Off-Dynamics Reinforcement Learning. NeurIPS 2024.

---

> > ### Author Rebuttal · Reviewer_WwvE · 2026-04-02
> >
> > On Weakness 1, my main concern was the gap between the discrete partition of heterogeneous sub-datasets in the theory and the continuous latent $z$ used in the proposed modality-aware advantage learning method. Section 4.2 and Answer 1 clarify the motivation more clearly, and I no longer view this as a blocking issue.
> >
> > The additional Adroit results address my concern in Weakness 2. For Weakness 3, I appreciate the authors' commitment to revising the highlighting in Table 2. Weakness 4 was not a major factor in my assessment.
> >
> > Question 1 has also been addressed. For Question 2, I had overlooked that the motivating example already combines heterogeneity in both dynamics and policy quality, consistent with the setting described in Definition 4.2. Answer 8 resolves that concern. I still think another plausible form of heterogeneity is to have different policies under the same dynamics; for example, both a kinematic and a kinematic-free dataset could each contain medium and expert data. That said, this seems compatible with the current framework by further partitioning the source dataset.
> >
> > Overall, the rebuttal addresses my main concerns and moves me toward a more positive assessment. I have therefore increased my significance score to 3 and my overall recommendation to 4.

---

> > > ### Author Response · Authors · 2026-04-02
> > >
> > > We are glad that our response has addressed the reviewer's major concerns. We deeply appreciate the reviewer's time dedicated to reviewing our paper.

---

### Official Review · Reviewer_uXVB · 2026-03-12

**Soundness:** 2
**Presentation:** 2
**Significance:** 2
**Originality:** 2
**Overall Recommendation:** 4
**Confidence:** 4

**Summary:**

This paper presents a cross-domain offline reinforcement learning in a general heterogeneous setting, where the source dataset is collected from multiple domains with varying dynamics and diverse behavior policies. Experiments on MuJoCo locomotion tasks validate the effectiveness of the proposed method.

**Compliance With Llm Reviewing Policy:**

Affirmed.

**Final Justification:**

The authors' rebuttal successfully addressed my initial concerns regarding the method's evaluation and scalability. The provided clarifications and additional details further validate the effectiveness of the proposed approach within the heterogeneous cross-domain offline RL setting.

**Key Questions For Authors:**

See the weaknesses.

**Limitations:**

The limitations of the proposed method should be included.

**Strengths And Weaknesses:**

**Strengths:**

- This paper introduces a more practical heterogeneous cross-domain offline RL setting, compared with the unimodal source dataset setting.
- The proposed V2A framework mitigates value misassignment through temporally consistent modality representation learning and modality-aware advantage learning.

**Weaknesses:**

- Beyond the relatively mild cross-dynamics setting, it would be interesting to examine whether the method can also handle more challenging settings, such as cross-task or cross-environment transfer.
- The effectiveness is validated only on the D4RL benchmark. It would strengthen the paper to evaluate the method on the OGBench [1] benchmark.
- Since the proposed framework introduces additional components, the paper should include the number of parameters comparison.
- Could the authors explain why the proposed method underperforms the baseline in certain cases?

References:

[1] Park et al. "OGBench: Benchmarking Offline Goal-Conditioned RL", ICLR, 2025.

---

> ### Author Rebuttal · Authors · 2026-03-28
>
> We thank Reviewer uXVB for the insightful comments. Below are our responses to the concerns.
>
> **1. More evaluations on cross-environment transfer**
>
> To address this concern, we consider a cross-environment setting where source and target domains have different state and action spaces.
>
> Following [1], we modify MuJoCo environments to construct source domains, including a three-legged halfcheetah and a five-legged ant. Following the same process as in Section 6.1, we build heterogeneous source datasets with medium-replay and medium-expert quality, while the target dataset is sampled from D4RL expert data.
>
> To address distinct state-action spaces, we first learn an inter-domain mapping via dynamics cycle consistency [1]. We then train IGDF, DVDF, and V2A and evaluate them on the target domain. As shown below, V2A still outperforms the baselines, verifying its effectiveness in cross-environment transfer.
>
> | | IGDF | DVDF | V2A |
> |-|-|-|-|
> |half-mr-e| 28.4 $\pm$ 2.3 | 31.5 $\pm$ 1.8 | **40.7 $\pm$ 3.2** |
> |half-me-e| 35.7 $\pm$ 3.1 | 40.2 $\pm$ 2.5 | **52.8 $\pm$ 4.1** |
> |ant-mr-e| 58.6 $\pm$ 3.0 | 53.1 $\pm$ 4.5 | **61.0 $\pm$ 3.5** |
> |ant-me-e| 82.3 $\pm$ 2.8 | 94.8 $\pm$ 1.5 | **103.3 $\pm$ 2.2** |
> |total| 205.0 | 219.6 | **257.8** |
>
> **2. More evaluations on other benchmarks**
>
> We thank the reviewer for suggesting OGBench. We respectfully clarify that OGBench is designed for goal-conditioned RL rather than cross-domain RL. It does not provide source-target paired domains, cross-domain baselines, or any cross-domain RL results for reference, making fair comparison significantly difficult.
>
> To address this concern, we instead follow ODRL [2], which is specifically developed for cross-domain RL and includes challenging domains such as Adroit. We additionally evaluate V2A on Adroit, a high-dimensional, long-horizon manipulation domain.
>
> Specifically, we consider Pen and Door. Following ODRL, we introduce easy/medium kinematic shifts (broken-joint) and morphology shifts (shrink-finger). For each shift, we collect medium-replay and medium-expert data and construct heterogeneous source datasets as in Section 6.1, while using 10% expert target data.
>
> We compare IGDF, DVDF, and V2A on these tasks. As shown below, V2A still outperforms baselines on these challenging Adroit tasks.
>
> | | IGDF | DVDF | V2A |
> |-|-|-|-|
> |Pen-mr-e| 12.2 $\pm$ 2.1 | 11.5 $\pm$ 1.8 | **17.3 $\pm$ 2.4** |
> |Pen-me-e| 20.5 $\pm$ 2.3 | 25.8 $\pm$ 1.9 | **38.7 $\pm$ 3.6** |
> |Door-mr-e| 35.3 $\pm$ 1.8 | 47.6 $\pm$ 2.0 | **54.4 $\pm$ 2.1** |
> |Door-me-e| 42.8 $\pm$ 4.5 | 48.2 $\pm$ 1.7 | **61.9 $\pm$ 2.3** |
> |total| 110.8 | 133.1 | **172.3** |
>
> **3. Including comparison of parameter numbers**
>
> We appreciate this suggestion. We compare the average parameter numbers of V2A, DVDF, IGDF, and OTDF across halfcheetah, hopper, walker2d, and ant. The increased parameter numbers for V2A mainly come from the additional representation-learning module.
>
> | V2A (IGDF) | DVDF (IGDF) | IGDF |  V2A (OTDF) | DVDF (OTDF) | OTDF |
> |-|-|-|-|-|-|
> |1.28M|0.88M|0.65M|1.34M|0.93M|0.70M|
>
> **4. Why V2A underperforms the baseline in certain cases**
>
> We believe there are two possible reasons. First, the learned representation $z$ is informative but not perfect. As shown in Fig. 2(a), the latent distributions across tasks are largely separated but still slightly overlapping, so value misassignment can be mitigated but not fully eliminated. Second, we fix the trade-off hyperparameter λ=0.6 across all datasets without task-specific tuning, and different datasets may prefer different λ values for the dynamics-value trade-off.
>
> To verify this, we additionally tune λ on several datasets where V2A underperforms the baselines. Note that when λ=1, V2A reduces to IGDF. As shown below, with a suitable λ, V2A can outperform the baselines on these tasks. We fix λ=0.6 in the paper because it already gives strong overall performance.
>
> | | λ=0.5 | λ=0.6 | λ=0.7 | λ=0.8 | λ=0.9 | IGDF (λ=1) | DVDF |
> |-|-|-|-|-|-|-|-|
> |half-mr-m| 22.2 $\pm$ 1.8 | 26.8 $\pm$ 2.4 | **29.5 $\pm$ 3.2** | **28.8 $\pm$ 1.9** | 27.3 $\pm$ 2.1 | 24.6 $\pm$ 2.1 | 28.3 $\pm$ 1.8 |
> |half-mr-e| 23.5 $\pm$ 3.1 | 26.4 $\pm$ 0.5 | **31.8 $\pm$ 2.0** | 27.6 $\pm$ 0.7 | 27.7 $\pm$ 1.2 | **28.2 $\pm$ 1.4** | 25.0 $\pm$ 1.2 |
> |walker2d-mr-me| 26.4 $\pm$ 3.2 | 31.4 $\pm$ 2.7 | 28.1 $\pm$ 1.8 | **36.7 $\pm$ 3.2** | **38.4 $\pm$ 2.0** | 35.2 $\pm$ 2.8 | 30.0 $\pm$ 1.4 |
> |ant-mr-e| 67.2 $\pm$ 3.3 | 70.2 $\pm$ 2.3 | 71.5 $\pm$ 2.8 | **74.0 $\pm$ 3.5** | **73.9 $\pm$ 4.2** | 73.1 $\pm$ 4.5 | 65.5 $\pm$ 3.0 |
>
>
> **5: Limitations should be included**
>
> We appreciate this suggestion. We will explicitly discuss the limitations of V2A in our updated version, including the extra time and parameters required for representation learning.
>
> [1] Learning cross-domain correspondence for control with dynamics cycle-consistency. ICLR 2021.
>
> [2] ODRL: A Benchmark for Off-Dynamics Reinforcement Learning. NeurIPS 2024.

---

> > ### Author Rebuttal · Reviewer_uXVB · 2026-04-01
> >
> > Thank you for your response. I will increase my score.

---

> > > ### Author Response · Authors · 2026-04-01
> > >
> > > We are pleased that our response has addressed the reviewer’s concerns. We sincerely appreciate the reviewer’s positive assessment of our work and the time dedicated to reviewing our paper.

---

### Official Review · Reviewer_f1UE · 2026-03-13

**Soundness:** 3
**Presentation:** 4
**Significance:** 4
**Originality:** 4
**Overall Recommendation:** 5
**Confidence:** 4

**Summary:**

This paper addresses heterogeneous cross-domain offline RL, where source data comes from multiple environments with differing dynamics and behavior policies. The authors identify a failure mode called 'value misassignment': prior methods train a global advantage function on pooled data, systematically underestimating the quality of source trajectories that are optimal under local dynamics but diverge from the global average. To solve this, they propose V2A, employing an EM-based framework with a Temporally-Consistent ELBO to infer trajectory-level dynamics modalities. The advantage function is conditioned on these modalities, ensuring accurate data filtering before target-policy optimization.

**Compliance With Llm Reviewing Policy:**

Affirmed.

**Final Justification:**

Thanks for the replies. I accept your point that dropping (z) from the advantage breaks the setup you need, the capacity-matched run is a fair way to separate modality-aware assignment from extra capacity in the advantage path. The continuous heterogeneous experiments help with my concern about only discrete, balanced mixes. Please state the theory assumptions modestly in the text and add the limitations you promised. DVDF alignment and code are still open for me. Still weak accept.

**Key Questions For Authors:**

1. Can you provide a controlled test where the trajectory representation $z$ is learned, but the advantage function is not conditioned on it (i.e., using $A(s,a)$ instead of $A(s,a,z)$ in Equation 8)? This isolation test is strictly necessary to cleanly isolate the effect of value assignment from the sheer capacity of the representation pre-training phase.

2. The theoretical proofs in Appendix B assume local convexity of $F(P)$ and that all component source dynamics lie in a close neighborhood. Do you believe these assumptions actually hold in your MuJoCo experiments, or are they strictly intended for theoretical motivation? Verifying these assumptions empirically would greatly strengthen the soundness of the paper.

3. How does V2A perform on highly skewed or imbalanced data mixtures (e.g., 90% one domain, 10% another), or on continuous parameter distributions, rather than the discrete, perfectly balanced 4-domain split used in the current experiments? Demonstrating robustness to imbalanced mixtures is crucial for claiming utility in 'in-the-wild' datasets.

**Limitations:**

The authors discuss computational time costs in the appendix but fail to address methodological limits. A proper limitations section must discuss the unverified assumptions of the theoretical bounds, the reliance on perfectly discrete and synthetic dynamics shifts, and the assumption that dynamics remain perfectly stationary within a single trajectory (which justifies the trajectory-level ELBO).

**Strengths And Weaknesses:**

The primary strength of the paper is its precise conceptual identification of the 'value misassignment' problem. The motivating example presented in Section 3 is exemplary. It empirically demonstrates exactly why global advantage estimation fails on heterogeneous data. By formalizing this intuition into a rigorous framework, the authors provide a highly transferable insight for the broader offline RL community. Proposition 4.5 provides a formal mathematical grounding for this phenomenon, elegantly decomposing the suboptimality bound into per-modality terms. The experimental execution is unusually thorough for the subfield, testing multiple target data qualities, various source shift types, and running across 5 random seeds to ensure statistical reliability. The t-SNE and advantage distribution plots in Figure 2 provide clear mechanistic evidence that the learned latent representations successfully separate distinct dynamics modes and sharpen the value estimation, fully supporting the theoretical claims.

The main weakness of the paper is its heavy reliance on entirely synthetic datasets. The heterogeneity is constructed by manually editing MuJoCo XML files and mixing them in perfectly balanced 25% proportions. While this is standard practice for initial benchmarking, this highly structured and discrete heterogeneity does not guarantee that the EM-based inference will succeed on the continuous, messy variations found in real-world 'in-the-wild' robotic datasets. Second, the paper lacks a definitive isolation experiment to completely isolate its core mechanism. Figure 4 tests the removal of the TC-ELBO, but the paper never tests a variant that keeps the latent encoder $z$ but removes it from the advantage function. Without this specific controlled test, it is difficult to cleanly separate the benefits of modality-aware assignment from the benefits of simply having a highly-parameterized representation pre-training phase.

The computational overhead introduced by V2A is massive. The method required steps of representation learning and the additional steps of advantage pre-training before the actual policy optimization even begins. This substantially increases the barrier to entry for practical deployment. The theoretical justification relies on strong assumptions that are neither surfaced in the main text nor verified empirically. Specifically, the proof of Proposition 4.5 assumes local convexity of the value gap function and requires that all source dynamics sit within a close neighborhood of the target, which may not hold true for the larger dynamics shifts tested in the experiments.

---

> ### Author Rebuttal · Authors · 2026-03-27
>
> We thank Reviewer f1UE for the insightful comments and the recognition of our work. Below are our responses to the raised concerns.
>
> **1. Controlled test for the effect of the representation**
>
> We greatly appreciate this suggestion. We respectfully clarify that learning $z$ without conditioning the advantage on it may not be an effective ablation. If $z$ is not used in the advantage function, it becomes causally disconnected from filtering and policy optimization process. In that case, the representation pre-training phase cannot affect the final performance. Note that Equations 6 and 7 are merely used to obtain the advantage function and are disentangled from the cross-domain policy optimization process.
>
> Instead, we conduct an alternative ablation study to examine whether the performance gain stems from introducing the additional input dimension of $z$, which in turn increases the capacity of the advantage network. Specifically, for the original
> $A(s,a)$ network, we add one extra hidden layer of size 256 to enlarge its capacity, while keeping the $A(s,a,z)$ network unchanged and all other training settings identical. We then use the original $A(s,a)$, the enlarged $A(s,a)$, and $A(s,a,z)$ respectively for data filtering and evaluate on multiple datasets. Results are reported in the table below. As shown, merely increasing the capacity of $A(s,a)$ does not yield a clear performance improvement, whereas $A(s,a,z)$ consistently achieves superior performance. This suggests that the performance gain comes from the representation $z$ alleviating value misassignment, rather than from increased network capacity.
>
> | | Original $A(s,a)$ | Enlarged $A(s,a)$ | $A(s,a,z)$ |
> |-|-|-|-|
> |half-me-e| 46.9 $\pm$ 2.1 | 48.2 $\pm$ 2.3 | **62.7 $\pm$ 4.8** |
> |hopper-me-e| 55.3 $\pm$ 4.4 | 55.7 $\pm$ 3.8 | **64.8 $\pm$ 3.6** |
> |walker2d-me-e| 67.1 $\pm$ 2.8 | 67.4 $\pm$ 1.2 | **79.4 $\pm$ 5.6** |
> |ant-me-e| 121.8 $\pm$ 0.5 | 122.3 $\pm$ 0.8 | **124.3 $\pm$ 0.2** |
> total| 291.1 | 293.6 | **331.2** |
>
> **2. On the assumptions of the theoretical analysis**
>
> We acknowledge that these assumptions are mainly introduced for theoretical analysis. Their empirical verification might be difficult as the true underlying transition models are unknown in practice. However, these assumptions are common and reasonable in theory.  The local convexity of $F(P)$ relies on a second-order sufficient condition, which is standard in numerical optimization. The "close neighborhood" assumption requires the source dynamics not to deviate too far from each other, which can be reasonable in controlled settings. We will provide further clarification in our updated version.
>
> **3. Evaluation on unstructured heterogeneous settings**
>
> Following the reviewer's advice, we additionally consider a heterogeneous setting with continuous parameter variations. Specifically, for each target environment (halfcheetah, hopper, walker2d, ant), we first train an expert policy on the target domain. When collecting the source dataset by rolling-out this expert policy, we randomly perturb the robot joint rotation angles within a certain range at the beginning of each episode via Python API, thereby constructing continuous kinematic shifts across source domains. Since the expert policy trained on the target domain can be suboptimal under these shifted source dynamics, the resulting data quality also varies continuously, yielding a heterogeneous source dataset with both continuous dynamics changes and continuous quality variations. The target dataset is composed of expert data.
>
> We compare IGDF, DVDF and V2A under this setting, with the results shown in the table below. As we can see, even in this less structured heterogeneous setting, V2A significantly outperforms IGDF and DVDF. We believe this is because the learned representation $z$ is continuous, and can therefore capture continuous variations in the underlying dynamics.
>
> | | IGDF | DVDF | V2A |
> |-|-|-|-|
> |halfcheetah| 38.2 $\pm$ 4.6 | 32.6 $\pm$ 2.8 | **53.7 $\pm$ 3.6** |
> |hopper| 43.3 $\pm$ 2.2 | 49.4 $\pm$ 3.1 | **60.7 $\pm$ 5.6** |
> |walker2d| 47.5 $\pm$ 3.8 | 52.6 $\pm$ 2.5 | **66.3 $\pm$ 3.9** |
> |ant| 73.5 $\pm$ 3.9 | 77.8 $\pm$ 4.5 | **92.4 $\pm$ 5.4** |
> total| 202.5 | 212.4 | **273.1** |
>
> **4. Undiscussed methodological limitations**
>
> We deeply appreciate this suggestion. We will add a dedicated limitation section in the updated version to discuss the underlying assumptions for our theory and TC-ELBO, as well as the experimental settings on synthetic dynamics shifts.

---

> > ### Author Rebuttal · Reviewer_f1UE · 2026-04-04
> >
> > Thanks for the replies. I accept your point that dropping (z) from the advantage breaks the setup you need, the capacity-matched run is a fair way to separate modality-aware assignment from extra capacity in the advantage path. The continuous heterogeneous experiments help with my concern about only discrete, balanced mixes. Please state the theory assumptions modestly in the text and add the limitations you promised. DVDF alignment and code are still open for me. Still weak accept.

---

> > > ### Author Response · Authors · 2026-04-04
> > >
> > > We are glad that our additional representation ablation and continuous heterogeneous experiments help address the reviewer’s major concerns. In the updated version, we will state the theoretical assumptions for Proposition 4.5 (the convexity assumption and source-domain similarity) and Proposition 5.1 (the stationary-dynamics assumption) more modestly, and include a limitation section in the main text to discuss the methodological limits of V2A, including the theoretical assumptions and the experimental settings.
> > >
> > > Regarding the remaining concern about DVDF alignment and code, we would like to clarify that **we contacted the DVDF authors for the code and also confirmed that an official implementation is now publicly available.** Therefore, our DVDF baseline is implemented based on the official codebase, and we believe the comparison in our paper is reliable and fair.

---

### Official Review · Reviewer_qf3s · 2026-03-13

**Soundness:** 3
**Presentation:** 2
**Significance:** 3
**Originality:** 2
**Overall Recommendation:** 4
**Confidence:** 3

**Summary:**

This paper proposes V2A (Value Alignment and Assignment), a framework for heterogeneous cross-domain offline reinforcement learning where source datasets may come from multiple domains with distinct dynamics and behavior policies. The paper identifies a problem termed "value misassignment" - where advantage values computed on mixed dynamics lead to incorrect sample optimality assessment - and proposes temporally-consistent modality representation learning plus modality-aware advantage learning to address it.

**Compliance With Llm Reviewing Policy:**

Affirmed.

**Final Justification:**

Raise to 4 since all concerns resolved.

**Key Questions For Authors:**

Please check the weakness.

**Limitations:**

(a) computational overhead from trajectory-level representation learning for long-horizon tasks, (b) settings where dynamics divergence is too large for modality disentanglement, or (c) whether V2A's gains diminish with more homogeneous source datasets.

**Strengths And Weaknesses:**

# Strengths

Practically Relevant Problem Setting: The paper addresses heterogeneous cross-domain offline RL, where source datasets originate from multiple domains with distinct dynamics and behavior policies. This setting better reflects real-world scenarios (e.g., robotic learning with data aggregated from multiple environments) compared to prior work assuming unimodal source datasets.

Theoretical Motivation: Proposition 4.5 provides a suboptimality bound decomposing the performance gap into per-source-domain value misalignment and dynamics misalignment terms, plus an estimation error term. This bound motivates the design principle of prioritizing samples from sub-domains with smaller misalignment.

Consistent Empirical Improvements: V2A demonstrates performance gains across 24 task configurations (4 environments × 3 target data qualities × 2 base algorithms). IGDF-based V2A outperforms IGDF and DVDF in 20/24 tasks, and OTDF-based V2A outperforms in 21/24 tasks. Total score improvements are 21.4% (IGDF) and 22.2% (OTDF).

Representation Quality Evidence: Figure 2(a) shows t-SNE visualizations where trajectories from the same source domain or shift type cluster together, suggesting the temporally-consistent encoder learns meaningful dynamics modality representations. Figure 2(b) shows V2A's advantage distribution is sharper than DVDF's, consistent with better optimality estimation.

Ablation Study Included: Appendix F.1 (Figure 4) compares V2A with temporally-consistent ELBO against sample-V2A with transition-level ELBO, showing the trajectory-level encoding improves performance in 7/8 tasks.

Computational Overhead Analysis: Appendix F.3 (Figure 6) provides time cost breakdowns showing V2A's additional overhead is concentrated in decoupled pretraining stages (representation extraction, advantage learning) that can be precomputed.

# Weakness

Motivating Example Setup Inconsistency: The experimental setup constructs the "source dataset" by mixing (a) 0.5M expert data sampled from the target domain and (b) 0.5M medium data from a source domain with dynamics shifts. This does not faithfully represent the claimed heterogeneous setting where source data comes from multiple source domains. Including target-domain data artificially simplifies the value assignment problem because target data is by definition dynamics-aligned.

Unproven Theoretical Approximation: The derivation of value misassignment relies on the approximation E[A_insrc^(s,a)] ≈ J_M_src(μ_src^i) - J_M_src(π_insrc^) for advantage functions computed on a mixture MDP. The manuscript does not prove this approximation holds, specify conditions for validity (e.g., small dynamics divergence between sub-domains), or provide counterexamples showing when it fails.

Incomplete Problem-Method Alignment: The paper claims temporally-consistent (trajectory-level) representation learning solves value misassignment, but does not explain the mechanistic link: why does having different z values for transitions within the same trajectory cause incorrect advantage estimation? The connection between temporal consistency and correct value assignment is asserted but not derived.

Causal Attribution Without Ablation: The 37% performance improvement (63 vs 46) in the motivating example is attributed solely to fixing value misassignment. However, no ablation isolates the value assignment component from representation quality improvements or optimization effects. Alternative explanations (e.g., trajectory-level encoding provides better features regardless of value assignment) are not ruled out.

Baseline Fairness Context: DVDF was originally designed for unimodal source datasets. Comparing DVDF's performance on heterogeneous datasets (where it was not designed to excel) against V2A (specifically designed for this setting) may create an unfair comparison. The paragraph does not acknowledge DVDF's original design scope.

---

> ### Author Rebuttal · Authors · 2026-03-27
>
> We thank Reviewer qf3s for the insightful comments. Below are our responses to the concerns.
>
> **1. Motivating Example Setup Inconsistency**
>
> We respectfully clarify that the example is consistent with the claimed heterogeneous setting: the source data contains multiple dynamics (zero & medium shifts) and behavior policies (expert & medium). Sampling from the target domain is only to simulate a very small (zero) dynamics shift. Importantly, value misassignment is not alleviated, as it is rooted in the multiplicity of co-existing dynamics modalities, regardless of alignment.
>
> To further address this concern, we consider a clearer heterogeneous setting, where the source dataset mixes 0.5M expert data from halfcheetah-kinematic-easy and 0.5M medium-quality data from halfcheetah-kinematic-medium. We train V2A and DVDF and evaluate them on the target domain over 5 seeds, as shown below. It is clear that V2A still outperforms DVDF.
>
> | | V2A | DVDF |
> |-|-|-|
> |Score| **60.8 $\pm$ 2.1** | 42.6 $\pm$ 3.4 |
>
> **2. Unproven Theoretical Approximation**
>
> The performance difference lemma gives
>
> $J(\mu^i\_{src}) - J(\pi^{i\star}\_{insrc}) = \mathbb{E}\_{(s,a)\sim d^{\mu^i\_{src}}}[A^\star\_{insrc}(s,a)],$
>
> where, in practice, the occupancy-measure expectation $\\mathbb{E} _ {(s,a)\\sim d^{\\mu^i _ {src}}}$ is not directly accessible. We approximate it by the empirical expectation $\\mathbb{E} _ {(s,a)\sim \mathcal D^i\_ {src}}$, a standard and natural choice depending on our assumption of sufficient source data coverage (rather than the dynamics divergence between sub-domains).
>
> **3. Incomplete Problem-Method Alignment**
>
> Temporal consistency is crucial for correct value assignment. Without it, transitions from the same trajectory may be assigned different $z$'s, artificially splitting them into distinct modalities, introducing noise into advantage learning. E.g., the same $(s,a)$ pair in a trajectory with different $z_1, z_2$ would yield inconsistent advantage estimates $A(s,a,z_1) \neq A(s,a,z_2)$, leading to incorrect learning. V2A resolves this by enforcing a shared trajectory-level $z$.
>
> **4. Causal Attribution Without Ablation**
>
> We clarify that value assignment and representation quality are inherently coupled, as value assignment in V2A directly relies on the dynamics information in $z$. They are not designed to be separable. Figures 2(a-b) show $z$ effectively captures modalities critical for advantage estimation.
>
> We still conduct an ablation study to isolate the effect of $z$ from network capacity increases. Specifically, we enlarge the original $A(s,a)$ by adding a hidden layer of size 256, while keeping $A(s,a,z)$ unchanged and all other settings identical. We then compare the original $A(s,a)$, the enlarged $A(s,a)$, and $A(s,a,z)$ on several datasets. As shown below, simply increasing the capacity of $A(s,a)$ leads to no clear gains, whereas $A(s,a,z)$ consistently performs better. This confirms that the improvement comes from $z$ alleviating value misassignment, not from increased network capacity.
>
> | | Original $A(s,a)$ | Enlarged $A(s,a)$ | $A(s,a,z)$ |
> |-|-|-|-|
> |half-me-e| 46.9 $\pm$ 2.1 | 48.2 $\pm$ 2.3 | **62.7 $\pm$ 4.8** |
> |hopper-me-e| 55.3 $\pm$ 4.4 | 55.7 $\pm$ 3.8 | **64.8 $\pm$ 3.6** |
> |walker2d-me-e| 67.1 $\pm$ 2.8 | 67.4 $\pm$ 1.2 | **79.4 $\pm$ 5.6** |
> |ant-me-e| 121.8 $\pm$ 0.5 | 122.3 $\pm$ 0.8 | **124.3 $\pm$ 0.2** |
> total| 291.1 | 293.6 | **331.2** |
>
> **5. Baseline Fairness Context**
>
> DVDF is designed for general cross-domain RL. Its prior evaluation focused on simpler unimodal settings but does not preclude application to the general heterogeneous setting. DVDF's worse performance simply means it failed to tackle value misassignment. V2A is developed specifically to address this issue. Thus, comparing them is essentially fair and is an ablation that validates our core contribution.
>
> **6. Computational overhead from trajectory-level representation learning for long-horizon tasks**
>
> The additional overhead is mainly incurred in the modality representation learning stage and is decoupled from policy optimization. As such, it can be precomputed, incurring no extra cost in subsequent policy learning. We will clarify this in the updated version.
>
> **7. Dynamics divergence might be too large for modality disentanglement**
>
> We respectfully clarify that larger dynamics divergence generally makes modalities easier to distinguish. Since V2A learns modality representations from trajectory dynamics patterns, increased divergence strengthens modality separation.
>
> **8. Whether V2A's gains diminish with more homogeneous source datasets.**
>
> V2A's gain may diminish in this case, as value misassignment primarily exists in heterogeneous settings. In the homogeneous case, modality-aware advantage becomes global advantage, and V2A reduces to DVDF. However, this reflects that V2A serves as a unified framework that encompasses DVDF as a special case, offering additional benefits when heterogeneity is present.

---

> > ### Author Rebuttal · Reviewer_qf3s · 2026-04-02
> >
> > The problem-method alignment (Weakness 3) explanation is still hand-wavy. The rebuttal argues that without temporal consistency, transitions from the same trajectory get different z values, introducing "noise." But this is a re-statement of the design choice, not a formal derivation showing why inconsistent z values lead to incorrect advantage estimation. The mechanistic link between temporal consistency and correct value assignment remains asserted rather than proven.
> >
> > The causal attribution ablation (Weakness 4) is improved but not fully convincing. The new ablation comparing original vs. enlarged network capacity is a welcome addition. However, it only rules out network capacity as a confound — it does not rule out that trajectory-level encoding provides better features in general (independent of value assignment). The authors acknowledge that value assignment and representation quality are "inherently coupled" and "not designed to be separable," which means the core claim that improvements come specifically from fixing value misassignment cannot be cleanly validated.

---

> > > ### Author Response · Authors · 2026-04-06
> > >
> > > We thank the reviewer for responding to our rebuttal. We address the remaining concerns below.
> > >
> > > **1. Problem-Method Alignment**
> > >
> > > We provide a more rigorous analysis of why temporal consistency is needed for correct value assignment.
> > >
> > > Suppose maximizing TC-ELBO yields a trajectory-level latent $z _ \\mathrm{src}\\sim q _ \\psi(\\cdot|\\tau)$ that captures the dynamics latent of the source MDP $\\mathcal{M} _ \\mathrm{src}$. Thus, $A^\\star _ \\mathrm{insrc}(s,a,z _ \\mathrm{src})\\approx A^{\\pi _ \\mathrm{insrc}^\\star} _ {\\mathcal{M} _ \\mathrm{src}}(s,a)$.
> > >
> > > Consider a sufficiently long source trajectory,
> > > $\\tau=\\{(s _ t,a _ t,r _ t,s _ {t+1})\\} _ {t=0}^{T-1}$, sampled from $\\mathcal{M} _ \\mathrm{src}$ under $\\mu _ \\mathrm{src}$. By the performance difference lemma,
> > >
> > > $\\mathbb{E} _ {(s,a,z _ \\mathrm{src})\\in\\tau}[A^\\star _ \\mathrm{insrc}(s,a,z _ \\mathrm{src})]\\approx\\mathbb{E} _ {(s,a)\\in\\tau}[A^{\\pi^\\star _ \\mathrm{insrc}} _ {\\mathcal{M} _ \\mathrm{src}}(s,a)]\\approx J _ {\\mathcal{M} _ \\mathrm{src}}(\\mu _ \\mathrm{src})-J _ {\\mathcal{M} _ \\mathrm{src}}(\\pi^\\star _ \\mathrm{insrc}),$
> > >
> > > where the second approximation holds by the sufficient data assumption.
> > >
> > > Note that $J _ {\\mathcal{M} _ \\mathrm{src}}(\\mu _ \\mathrm{src})-J _ {\\mathcal{M} _ \\mathrm{src}}(\\pi^\\star _ \\mathrm{insrc})$ is the correct value misalignment term in Proposition 4.5. Thus, no additional value misassignment is introduced.
> > >
> > > Without temporal consistency, each transition in $\\tau=\\{(s _ t,a _ t,r _ t,s _ {t+1})\\} _ {t=0}^{T-1}$ is assigned an individual latent $\\{z _ t\\sim q _ \\psi(\\cdot|s _ t,a _ t,s _ {t+1})\\} _ {t=0}^{T-1}$. For analysis, we discretize the latent space into $K$ dynamics modes, which correspond to $K$ different MDPs: $\\{\\mathcal{M} _ 1,...,\\mathcal{M} _ K\\}$. Then $\\tau$ is partitioned into $K$ transition subsets $\\{\\mathcal{T} _ 1,...,\\mathcal{T} _ K\\}$ where
> > >
> > > $\\mathcal{T} _ j=\\{(s _ t,a _ t,r _ t,s _ {t+1})\\in\\tau|z _ t \\, \\text{corresponds to \\,} \\mathcal{M} _ j\\}, \\qquad j=1,2,...,K$
> > >
> > > For each transition subset $\\mathcal{T} _ j$, we have $A^\\star _ \\mathrm{insrc}(s _ t,a _ t,z _ t)\approx A^{\\pi _ \\mathrm{insrc}^{j,\\star}} _ {\\mathcal{M} _ j}(s _ t,a _ t)$. Hence,
> > >
> > > $\\mathbb{E} _ {(s _ t,a _ t,s _ {t+1})\\in\\tau,z _ t\\sim q _ \\psi(\\cdot|s _ t,a _ t,s _ {t+1})}[A^\\star _ \\mathrm{insrc}(s _ t,a _ t,z _ t)]\\approx \\sum _ {j=1}^K \\frac{|\\mathcal{T} _ j|}{|\\tau|}\\mathbb{E} _ {(s _ t,a _ t)\\in\\mathcal{T} _ j}[A^{\\pi _ \\mathrm{insrc}^{j,\star}} _ {\\mathcal{M} _ j}(s _ t,a _t )].$
> > >
> > > Applying the performance difference lemma to each $\\mathcal{T} _ j$, we have:
> > >
> > > $\\mathbb{E} _ {(s _ t,a _ t)\\in\\mathcal{T} _ j}[A^{\\pi _ \\mathrm{insrc}^{j,\\star}} _ {\\mathcal{M} _ j}(s _ t,a _ t)]\\approx J _ {\\mathcal{M} _ j}(\\mu _ \\mathrm{src})-J _ {\\mathcal{M} _ j}(\\pi^{j,\\star} _ \\mathrm{insrc}).$
> > >
> > > where the approximation again uses the sufficient-data assumption. Therefore,
> > >
> > > $\\mathbb{E} _ {(s _ t,a _ t,s _ {t+1})\\in\\tau,z _ t\\sim q _ \\psi(\\cdot|s _ t,a _ t,s _ {t+1})}[A^\\star _ \\mathrm{insrc}(s _ t,a _ t,z _ t)]\\approx \\sum _ {j=1}^K \\frac{|\\mathcal{T} _ j|}{|\\tau|}(J _ {\\mathcal{M} _ j}(\\mu _ \\mathrm{src})-J _ {\\mathcal{M} _ j}(\\pi^{j,\\star} _ \\mathrm{insrc})).$
> > >
> > > In general, $\\sum _ {j=1}^K \\frac{|\\mathcal{T} _ j|}{|\\tau|}(J _ {\\mathcal{M} _ j}(\\mu _ \\mathrm{src})-J _ {\\mathcal{M} _ j}(\\pi^{j,\\star} _ \\mathrm{insrc})) \\neq J _ {\\mathcal{M} _ \\mathrm{src}}(\\mu _ \\mathrm{src})-J _ {\\mathcal{M} _ \\mathrm{src}}(\\pi^{\\star} _ \\mathrm{insrc})$. Thus, without temporal consistency, the learned advantage cannot represent the correct value misalignment term, so the value misassignment issue remains.
> > >
> > > **2. The causal attribution ablation**
> > >
> > > To further address this concern, we learn a behavior-policy representation which contains no dynamics information for value assignment. Specifically, we optimize:
> > >
> > > $\\texttt{TC-ELBO} _ {\\psi,\\theta}:=\\mathbb{E} _ {\\tau\\sim\\mathcal{D} _ \\mathrm{src},z\\sim q _ \\psi(\\cdot|\\tau)}\\left[\\sum _ {t=0}^{T-1}\\log p _ \\theta(a _ t|s _ t,z)-D _ {KL}(q _ \\psi(\\cdot|\\tau),p(\\cdot))\\right],$
> > >
> > > where $q _ \\psi$ is the trajectory-level representation encoder, and $p _ \\theta$ is the transition-level behavior policy decoder, $p(z)=\\mathcal{N}(0,I)$. We then construct $A(s,a,z)$ as in V2A. Note that $A(s,a,z)$ is behavior-policy-aware instead of dynamics-aware. We compare original V2A with this variant, termed behavior-V2A, on several tasks. As shown below, V2A consistently outperforms behavior-V2A. This indicates that the gain of V2A does not come merely from better representations, but from using dynamics-aware representations directly relevant to value assignment.
> > >
> > > | |V2A|behavior-V2A|
> > > |-|-|-|
> > > |half-me-e|**62.7±4.8**|44.7±3.3|
> > > |hopper-me-e|**64.8±3.6**|57.2±2.9|
> > > |walker2d-me-e|**79.4±5.6**|68.0±4.6|
> > > |ant-me-e|**124.3±0.2**|119.4±1.3|
> > > |total|**331.2**|289.3|

---

### Decision · Program_Chairs · 2026-04-30

**Decision:**

Accept (regular)

**Comment:**

This paper studies heterogeneous cross-domain offline RL and identifies a failure mode termed value misassignment, arising from applying a global advantage function to mixed-domain data. The proposed V2A framework addresses this via temporally-consistent representation learning and modality-aware advantage estimation.

Reviewers generally agree that the problem is important and the formulation of value misassignment is insightful. The method is technically sound and demonstrates consistent empirical improvements across a range of tasks. After the rebuttal, concerns regarding evaluation breadth and experimental setup are largely addressed.

The main remaining concerns include the incomplete mechanistic justification linking temporal consistency to correct value assignment, and the difficulty of fully isolating the contribution of value assignment from representation learning (qf3s). However, the additional analysis and ablations provided in the rebuttal make a reasonable case, and these limitations are not uncommon in this line of work.

Overall, the paper provides a meaningful contribution to heterogeneous offline RL, and reviewer consensus converges toward acceptance after rebuttal.